# CoIN: A Benchmark of Continual Instruction Tuning for Multimodal Large Language Models

Cheng Chen[1]    Junchen Zhu[2]    Xu Luo[2]    Heng Tao Shen[2,3]    Jingkuan Song[1]    Lianli Gao[1,*]

[1]Shenzhen Institute for Advanced Study, University of Electronic Science and Technology of China
[2]University of Electronic Science and Technology of China [3]Tongji University

## Abstract

Instruction tuning demonstrates impressive performance in adapting Multimodal Large Language Models (MLLMs) to follow task instructions and improve generalization ability. By extending tuning across diverse tasks, MLLMs can further enhance their understanding of world knowledge and instruction intent. However, continual instruction tuning has been largely overlooked and there are no public benchmarks available. In this paper, we present CoIN, a comprehensive benchmark tailored for assessing the behavior of existing MLLMs under continual instruction tuning. CoIN comprises 10 meticulously crafted datasets spanning 8 tasks, ensuring diversity and serving as a robust evaluation framework to assess crucial aspects of continual instruction tuning, such as task order, instruction diversity and volume. Additionally, apart from traditional evaluation, we design another LLM-based metric to assess the knowledge preserved within MLLMs for reasoning. Following an in-depth evaluation of several MLLMs, we demonstrate that they still suffer catastrophic forgetting, and the failure in instruction alignment assumes the main responsibility, instead of reasoning knowledge forgetting. To this end, we introduce MoELoRA which is effective in retaining the previous instruction alignment. Codes and datasets are publicly available https://github.com/zackschen/CoIN.

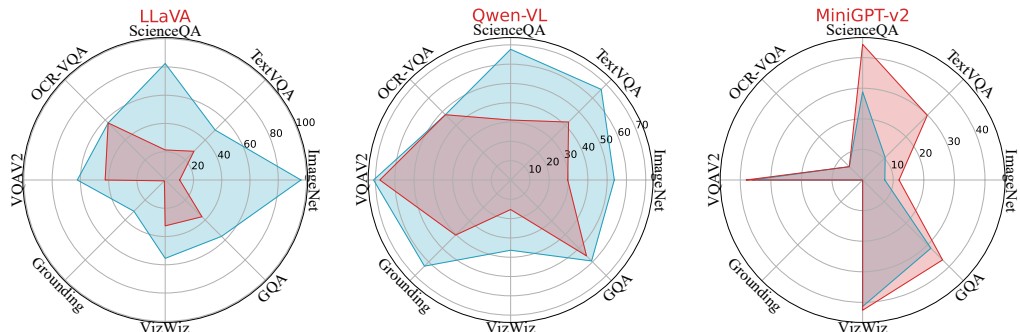

Figure 1: Different behavior of MLLMs when sequentially tuned on CoIN. Blue represents the accuracy for each task evaluated when just tuned on the corresponding task, and Red represents the accuracy evaluated after the models have been sequentially tuned on all tasks. LLaVA [35] and Qwen-VL [3] suffer from catastrophic forgetting while MiniGPT-v2 [9] does not. The sequential training starts clockwise from ScienceQA and ends with OCR-VQA.

38th Conference on Neural Information Processing Systems (NeurIPS 2024) Track on Datasets and Benchmarks.

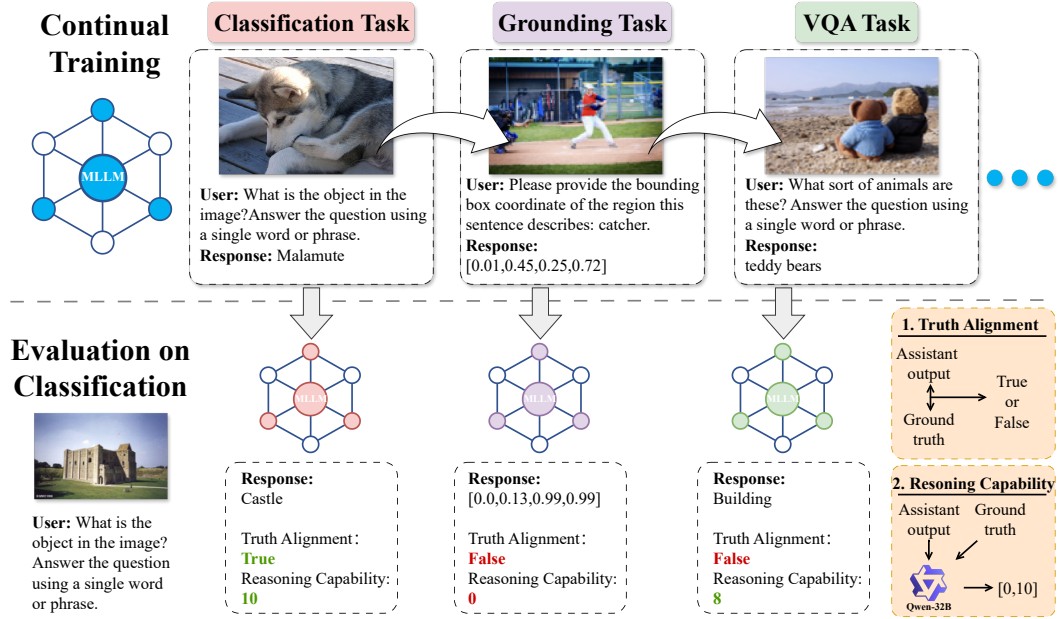

Figure 2: An overview of CoIN benchmark. A selected MLLM is sequentially fine-tuned on 8 instruction datasets spanning diverse tasks. Then, it is evaluated from two perspectives: *Truth Alignment* and *Reasoning Capability*, which assess the alignment with ground truth and knowledge preserved for reasoning, respectively. The evaluation example at the bottom presents the results of the model tested on classification after fine-tuning on each task.

# 1 Introduction

Recently, Multimodal Large Language Models (MLLMs) [8, 11, 35, 34, 70, 57, 40] have garnered significant attention for their remarkable capabilities of vision-language understanding and generation. These MLLMs commonly adopt two stages to learn extensive knowledge and align with different task instructions. In the initial stage, various pre-train strategies are employed to establish vision-language alignment. Subsequently, to enhance the capacity to follow task instructions and improve performance, the aligned MLLMs are fine-tuned on meticulously constructed instruction data.

Given the impressive performance of instruction tuning, researchers can further enhance the capacity of MLLMs to align with various task instructions and learn more world knowledge, by tuning across diverse tasks. However, the performance of previous tasks after sequential updating on different tasks has been largely overlooked. Recent years, continual learning (CL) [52, 2, 7] is proposed to investigate the behavior of artificial intelligence on sequential fine-tuning. Some research has delved into continual instruction tuning for Large Language Models (LLMs) [68, 61, 42, 31]. However, the exploration of continual instruction tuning for MLLMs has been overlooked. EMT [67] investigates the catastrophic forgetting of MLLMs, yet only focusing on classification problems, limiting the exploration of the diverse capabilities of powerful MLLMs. He et al. [21] propose a benchmark to explore whether multi-task joint instruction tuning enhances a model's continual learning ability. However, this setting limits the exploration of MLLMs in practical scenarios.

Therefore, to comprehensively investigate the behavior of MLLMs in continual instruction tuning, we introduce a novel benchmark: **Co**ntinual **I**nstruction tu**N**ing (CoIN). In CoIN, we construct a varied set of instructions by utilizing commonly used vision-language datasets to ensure accessibility and diversity, including general visual question answering, knowledge-grounded image question answering, OCR image question answering tasks, *etc*. However, the instructions constructed with these tasks mainly consist of question-answering tasks, lacking diversity. So, we additionally add grounding and classification tasks in our CoIN. Then, following common instruction templates [34, 35], we transform the selected datasets into a data format of instruction tuning.

Then, following an in-depth evaluation of popular MLLMs, we reveal that some of them still suffer from catastrophic forgetting (as showing in Fig. 1), similar to traditional continual CNN [22, 30]

---

*Corresponding author

or VIT [13, 38] models. However, different from these models which learn representations and use a final layer to make predictions with a fixed output style, MLLMs work in a generative way. This motivates us to inquire whether MLLMs forget the knowledge required for reasoning or if the issue lies in their inability to follow instructions. Because instruction tuning primarily focuses on learning to align with task instructions [69, 35, 34], we hypothesize that the model mainly loses the capability of instruction following, rather than the maintained knowledge. To validate this hypothesis, in addition to checking the outputs with the ground truth, called *Truth Alignment*, we employ powerful LLM assistant for evaluating the reasoning knowledge, called *Reasoning Capability*, as shown in Fig. 2.

After analyzing the results of these two evaluations, we reveal that the failure in instruction following assumes the main responsibility, instead of reasoning knowledge forgetting. Recently, Mixture-of-Experts (MoE) framework [54] leverages multiple distinct experts to acquire different knowledge and incorporates a gate function to modulate their contributions. We observe that this method resembles the architecture-based methods in traditional continual learning, providing the model with the ability to learn different instruction following from distinct experts. Therefore, we try to bring it into CoIN to mitigate the forgetting of alignments. Experimental results consistently demonstrate improvement after integrating with more experts.

In summary, the contributions of this paper are as follows:

- A novel benchmark for MLLMs in continual instruction tuning is proposed, namely CoIN, which consists of 10 datasets spanning 8 different tasks for comprehensive and diverse evaluation.

- A novel evaluation approach is introduced to assess the model's ability from two aspects: *Truth Alignment* and *Reasoning Capability*. Furthermore, we reveal that the catastrophic forgetting in MLLMs is primarily due to the decline in instruction following rather than reasoning knowledge.

- Multiple state-of-the-art MLLMs are chosen for evaluation on CoIN. Additionally, we introduce MoELoRA which is effective in mitigating forgetting owing to its use of distinct experts.

## 2    Preliminaries

In the multimodal continual instruction tuning, an MLLM has been pre-trained with abundant vision-language data to align the gap of vision and language, indicated by trainable parameters $\theta$. We further train it to adapt to novel $S$ tasks in a sequential manner. Each task is denoted by a task descriptor $\tau \in \{1, 2, ..., S\}$, and owns an independent dataset $D_\tau = \{(X_{\tau,j}^{img}, X_{\tau,j}^{ins}, X_{\tau,j}^{ans})_{j=1}^{N_\tau}\}$ with $N_\tau$ data pairs, where $X^{img}$, $X^{ins}$ and $X^{ans}$ indicate the input image tokens, instruction tokens and answer tokens, respectively. For a sample pair with an answer in length $L$, we compute the probability of the whole target answers $X^{ans}$ in an auto-regressive manner as follows:

$$p(X^{ans}|X^{img}, X^{ins}) = \prod_{i=1}^{L} p_\theta(X_i^{ans}|X^{img}, X^{ins}, X_{<i}^{ans}), \tag{1}$$

where $X_{<i}^{ans}$ indicates all answer tokens before the index $i$ and $X_i^{ans}$ indicates the $i$-th answer token. We optimize the network with the following function:

$$\mathcal{L} = -\sum_{i=1}^{L} \log p_\theta(X_i^{ans}|X^{img}, X^{ins}, X_{<i}^{ans}). \tag{2}$$

## 3    CoIN: A Benchmark for MLLMs

### 3.1    How to Compose A Comprehensive Benchmark for MLLMs?

#### 3.1.1    Data Integration.

A comprehensive evaluation relies on extensive data. To ensure the accessibility and diversity of the instruction tuning samples, we collect various publicly available and commonly used vision-language datasets. These datasets cover a wide range of tasks, including general image question answering, visual reasoning, knowledge-grounded image question answering, *etc*. Specifically, the selected datasets include VQAv2 [17], VizWiz [20], ScienceQA [41], TextVQA [55], GQA [26] and

Table 1: The statistic of collected datasets and instructions in CoIN benchmark.

| Task | Dataset | Instruction | Train Number | Test Number |
|---|---|---|---|---|
| Grounding | RefCOCO RefCOCO+ RefCOCOg | Please provide the bounding box coordinate of the region this sentence describes: <description> | 55k | 31k |
| Classification | ImageNet | What is the object in the image? Answer the question using a single word or phrase | 129k | 5k |
| Image Question Answering (IQA) | VQAv2 | Answer the question using a single word or phrase | 82k | 107k |
| Knowledge Grounded IQA | ScienceQA | Answer with the option's letter from the given choices directly | 12k | 4k |
| Reading Comprehension IQA | TextVQA | Answer the question using a single word or phrase | 34k | 5k |
| Visual Reasoning IQA | GQA | Answer the question using a single word or phrase | 72k | 1k |
| Blind People IQA | VizWiz | Answer the question using a single word or phrase | 20k | 8k |
| OCR IQA | OCR-VQA | Answer the question using a single word or phrase | 165k | 100k |

OCR-VQA [45]. However, we observe that these datasets are limited to traditional QA tasks in the vision-language community, lacking task diversity. To overcome this limitation, we introduce the classification task and region-level grounding task into CoIN with ImageNet [12], RefCOCO [28], RefCOCO+ [44] and RefCOCOg [44].

With a substantial collection of instruction data, the next step involves transforming these samples into a unified instruction tuning format. Nowadays, several works [34, 35, 11, 70] have resorted to different ways to construct the instructions. For example, LLaVA [35] leverages ChatGPT [46] and GPT-4 [47] to create GPT-assisted visual instruction based on COCO [33]. SPHINX [34] adapt different templates to transform a wide range of multi-modal tasks into instructions. Drawing on insights from prior research, we utilize commonly employed instruction templates to formulate our instructions, as illustrated in Tab. 1. The final benchmark encompasses 10 datasets spanning 8 task categories. (Some examples of our CoIN are presented in the Appendix).

### 3.1.2 Training

Low-rank Adaptation (LoRA) [25] has demonstrated effectiveness and efficiency in the fine-tuning of pre-trained language models. Additionally, a previous study [61] observed that parameter-efficient methods are more susceptible to forgetting in LLMs. Therefore, CoIN specifically explores the behavior of parameter-efficient fine-tuning of MLLMs, focusing on LoRA. Specifically, during fine-tuning on CoIN, only the low-rank matrices are updated, while the parameters of the base LLMs and vision encoder are frozen.

### 3.1.3 Performance Evaluation.

Different from traditional continual learning which only compares the predicted results with the ground truth in a word-for-word manner, *i.e. Truth Alignment*, we argue that the outputs of MLLMs are influenced by *Reasoning Capability*. Therefore, the evaluation of MLLMs needs to consider the two aspects respectively.

***Truth Alignment.*** The ability to generate the correct result in the desired format to follow task instruction is the basic requirement for instruction tuning. To evaluate the ability of overall performance of MLLMs on CoIN, we adopt the traditional evaluation method that we directly compare the outputs of MLLMs with ground truths. These metrics for each task slightly vary since different tasks have different forms of outputs (Comparison details can be found in the Appendix).

***Reasoning Capability.*** The performance of MLLMs depends not only on the instruction following but also on the knowledge maintained in MLLMs. For example, MLLMs may correctly answer the question logically as "Two apples" while the ground truth is "Two". The evaluation of the truth alignment will directly discriminate this sample as negative. To tackle this issue and further analyze

Table 2: The results evaluating the *Truth Alignment* ability are presented below. The first line of **Sequential Finetune** are the results for each task evaluated when just tuned on the corresponding task, and the second line displays the final results of each task after fine-tuning on the last task.

| MLLM | Method | Accuracy on Each Task | | | | | | | | Overall Results | |
|---|---|---|---|---|---|---|---|---|---|---|---|
| | | ScienceQA | TextVQA | ImageNet | GQA | VizWiz | Grounding | VQAV2 | OCR-VQA | MAA | BWT |
| LLaVA | Multi-task | 56.77 | 49.35 | 95.55 | 56.65 | 53.90 | 30.09 | 59.50 | 55.65 | 57.18 | - |
| | Zero-shot | 49.91 | 2.88 | 0.33 | 2.08 | 0.90 | 0.00 | 0.68 | 0.17 | 7.12 | - |
| | Sequential Finetune | 82.45 | 49.99 | 96.05 | 56.40 | 55.45 | 31.27 | 62.20 | 57.08 | 32.97 | -32.62 |
| | | 21.26 | 28.74 | 10.25 | 36.78 | 32.45 | 0.83 | 42.50 | 57.08 | | |
| Qwen-VL | Multi-task | 25.70 | 60.88 | 17.05 | 56.77 | 35.58 | 6.78 | 68.67 | 63.50 | 41.87 | - |
| | Zero-shot | 64.56 | 48.15 | 11.82 | 44.50 | 9.57 | 0.00 | 64.10 | 27.50 | 33.78 | - |
| | Sequential Finetune | 67.69 | 66.36 | 53.70 | 59.30 | 36.38 | 63.10 | 71.00 | 47.80 | 43.35 | -16.94 |
| | | 31.05 | 42.45 | 29.57 | 55.57 | 15.30 | 40.33 | 67.75 | 47.80 | | |
| MiniGPT-v2 | Multi-task | 43.55 | 19.24 | 10.57 | 28.43 | 41.62 | 0.00 | 27.12 | 1.45 | 21.50 | - |
| | Zero-shot | 32.16 | 6.83 | 0.07 | 11.58 | 35.20 | 0.00 | 12.20 | 0.03 | 12.26 | - |
| | Sequential Finetune | 28.81 | 10.40 | 7.25 | 31.55 | 41.35 | 0.00 | 36.10 | 6.15 | 25.45 | 6.04 |
| | | 44.35 | 29.89 | 11.90 | 36.95 | 42.58 | 0.00 | 38.10 | 6.15 | | |

MLLMs, we propose reasoning capability which refers to the knowledge contained in MLLMs to comprehend different modalities and make the reasoning. Motivated by previous works [15, 18], we adopt another LLM to grade the output. With designed prompts, the LLM will disregard the structure of the outputs and solely evaluate the key information within it to obtain a score from 0 to 10.

Roughly speaking, for an MLLM to generate a desired output, it must have the ability to make reasoning and transfer the reasoned output into a structure that aligns with task instructions. The *Truth Alignment* evaluates the overall performance of the model, while the *Reasoning Capability* specifically investigates the model's reasoning ability. The remaining aspect is the capability of *Instruction Following*. Therefore, the correlation of the above evaluations are following:

*Truth Alignment = Reasoning Capability + Instruction Following*

With the novel evaluations described above, we present the overall performance calculation in CoIN here. Adhering to traditional continual learning metrics, we employ Backward Transfer (BWT) to measure the degree of suffering catastrophic forgetting. Additionally, unlike traditional continual learning where the model gradually forgets learned knowledge, sharp fluctuations occur in instruction tuning influenced by the gap between different tasks. Hence, we incorporate an additional metric, Mean Average Accuracy [48], to measure the performance throughout the training process.

(1) *Mean Average Accuracy* (MAA): $MAA = \frac{1}{T}\sum_{j=1}^{T}(\frac{1}{j}\sum_{i=1}^{j}A_{j,i})$, where $A_{j,i}$ is the performance on $i$-th task after training the task $j$. A high MAA corresponds to a continual learning model that consistently maintains a high accuracy throughout the training process.

(2) *Backward Transfer* (BWT): $BWT = \frac{1}{T}\sum_{i=1}^{T}(A_{T,i} - A_{i,i})$, where $A_{i,i}$ is the performance on $i$-th task after training on $i$-th task.

# 4   Experiments

**Setup**   LLaVA [35], Qwen-VL [3] and MiniGPT-v2 [9] that have achieved remarkable performance on numerous benchmarks, are selected as the models for fine-tuning on the proposed CoIN benchmark. In addition, we choose two baselines for comparison: **Multi-task** which fine-tunes on all instructions instead of sequential training, and **Zero-shot** which involves assessing each task based on pre-trained MLLMs. For the fine-tuning sequence in CoIN, we adopt a random order, resulting in the following sequence: ScienceQA, TextVQA, ImageNet, GQA, VizWiz, Grounding, VQAv2, and OCR-VQA. For reasoning capability evaluation, we select Qwen-1.5-32B, a state-of-the-art model on many benchmarks, as the powerful LLM to evaluate the outputs from our trained model.

Table 3: The evaluation results of *Reasoning Capability* are presented below.

| MLLM | Method | Accuracy on Each Task | | | | | | | | Overall Results | |
|---|---|---|---|---|---|---|---|---|---|---|---|
| | | ScienceQA | TextVQA | ImageNet | GQA | VizWiz | Grounding | VQAV2 | OCR-VQA | MAA | BWT |
| LLaVA | Multi-task | 80 | 75 | 97 | 72 | 42 | 86 | 73 | 79 | 75.50 | - |
| | Zero-shot | 93 | 83 | 69 | 64 | 48 | 35 | 64 | 66 | 65.25 | - |
| | Sequential Finetune | 92 | 75 | 97 | 72 | 42 | 58 | 75 | 78 | 71.28 | -10.88 |
| | | 82 | 74 | 55 | 56 | 47 | 52 | 58 | 78 | | |
| Qwen-VL | Multi-task | 98 | 82 | 68 | 77 | 50 | 51 | 82 | 88 | 74.50 | - |
| | Zero-shot | 97 | 81 | 78 | 74 | 54 | 58 | 81 | 74 | 74.63 | - |
| | Sequential Finetune | 96 | 83 | 86 | 78 | 51 | 82 | 82 | 75 | 80.97 | -3.25 |
| | | 95 | 78 | 77 | 77 | 47 | 76 | 82 | 75 | | |
| MiniGPT-v2 | Multi-task | 96 | 76 | 58 | 62 | 44 | 89 | 63 | 59 | 68.38 | - |
| | Zero-shot | 98 | 72 | 48 | 63 | 48 | 80 | 64 | 61 | 66.75 | - |
| | Sequential Finetune | 97 | 71 | 55 | 61 | 44 | 91 | 63 | 52 | 75.05 | 0.00 |
| | | 89 | 73 | 59 | 60 | 44 | 94 | 63 | 52 | | |

## 4.1 How Do Existing MLLMs Perform in CoIN?

To comprehensively investigate the performance of the chosen MLLMs, we conduct experiments on our CoIN. Quantitative results about the ability of *Truth Alignment* and *Reasoning Capability* are shown in Tab. 2 and Tab. 3, respectively.

For the results of truth alignment of Tab. 2, we have the following observations: **Firstly**, unlike traditional continual learning, where the multi-task model often serves as the upper bound, in CoIN, the performance of the multi-task model is not the best due to the influence of task gaps. **Secondly**, even though pre-trained MLLMs retain substantial knowledge, the performance of zero-shot on specific tasks remains unsatisfactory, resulting in an accuracy of 7.12, 33.78 and 12.26. This validates the importance of instruction tuning for MLLMs in achieving task alignment. **Thirdly**, sequential finetune even performs better on fine-tuning tasks than multi-task (*i.e.* the first line in Sequential Finetune), except for some tasks in MiniGPT-v2. The possible reason may be that the model tends to focus on one task, diminishing the impact of diverse instructions from other tasks. However, due to the absence of techniques to regulate learning, these models suffer from forgetting, resulting in -32.62 of LLaVA and -16.94 of Qwen-VL in terms of BWT. **Finally**, MiniGPT-v2 demonstrates an incredible ability to mitigate forgetting. However, compared to LLaVA and Qwen-VL, it behaves underfitting on some tasks during fine-tuning. As training progresses, the performance on each task gradually improves. We believe this may be due to fewer training samples and iterations compared to its official instruction tuning.

In addition, comparing the results of *Truth Alignment* with those of *Reasoning Capability* in Tab. 3, it is evident that the forgetting of reasoning knowledge is much smaller than that of truth alignment. For example, the *Reasoning Capability* of LLaVA for grounding only drops from 58% to 52%, whereas the *Truth Alignment* drops from 31.27% to 0.00%. This comparison supports the hypothesis that the model primarily loses the capability to align with task instruction, rather than the maintained knowledge. Furthermore, on the one hand, compared with the slight decrease observed in LLaVA and Qwen-VL, MiniGPT-v2 performs robustly in retaining reasoning knowledge. On the other hand, compared to the truth alignment of MiniGPT-v2, it is noticeable that the overall performance increase of MiniGPT-v2 is primarily owing to the learning of instruction following. This coincides with the purpose of instruction tuning, which is to learn to align with task instruction. Finally, since the *Reasoning Capability* is robust to forgetting, we will just record the results of *Truth Alignment* in the following experiments (More results about *Reasoning Capability* are in Appendix).

## 4.2 Whether is Qwen a good evaluator?

We select Qwen to evaluate the *Reasoning Capability*, but is it a reliable evaluator? To assess its effectiveness, we conduct experiments to compare with another powerful closed-source large language model, along with a user study. The comparison results are presented below. Firstly, many works [18, 37] have commonly employed GPT-4 to evaluate the quality of generated samples. Following this approach, we also use GPT-4 to assess outputs using the same prompts. The comparison with Qwen reveals that the overall trends in evaluating *Reasoning Capability* are consistent. Secondly, we randomly sample model outputs for each task and gather feedback from AI researchers, asking them

to score the outputs by using the same prompts with both GPT-4 and Qwen-32B. The results from the user study align closely with those of Qwen-32B, confirming its validity as a reliable evaluator. In summary, Qwen is effective in assessing the retention and forgetting of *Reasoning Capability*.

Table 4: The comparison of Qwen with GPT-4 and user study as a evaluator are presented below.

| Type | Accuracy on Each Task | | | | | | | | Overall Results | |
|------|-----------|---------|----------|-----|--------|-----------|-------|---------|-----|------|
|  | ScienceQA | TextVQA | ImageNet | GQA | VizWiz | Grounding | VQAV2 | OCR-VQA | MAA | BWT |
| Qwen-32B | 92 | 75 | 97 | 72 | 42 | 58 | 75 | 78 | 71.28 | -10.88 |
|  | 82 | 74 | 55 | 56 | 47 | 52 | 58 | 78 | | |
| GPT-4 | 94 | 83 | 96 | 83 | 79 | 71 | 81 | 69 | 73.62 | -11.50 |
|  | 80 | 83 | 65 | 67 | 62 | 70 | 68 | 69 | | |
| User Study | 96 | 82 | 98 | 85 | 80 | 65 | 86 | 70 | 74.35 | -8.13 |
|  | 85 | 80 | 85 | 71 | 76 | 57 | 73 | 70 | | |

## 4.3 What factors affect the performance?

Table 5: The results of LLaVA about **different task orders** are presented below.

| Order | Accuracy on Each Task | | | | | | | | Overall Results | |
|-------|-----------|---------|----------|------|--------|-----------|---------|---------|-----|------|
|  | ScienceQA | TextVQA | ImageNet | GQA | VizWiz | Grounding | VQAV2 | OCR-VQA | MAA | BWT |
| Random | 82.45 | 49.99 | 96.05 | 56.40 | 55.45 | 31.27 | 62.20 | 57.08 | 32.97 | -32.62 |
|  | 21.26 | 28.74 | 10.25 | 36.78 | 32.45 | 0.83 | 42.50 | 57.08 | | |
|  | GQA | Grounding | ImageNet | OCR-VQA | ScienceQA | TextVQA | VizWiz | VQAV2 | MAA | BWT |
| Alphabet | 62.68 | 37.73 | 97.30 | 62.00 | 59.98 | 50.98 | 60.10 | 67.28 | 31.08 | -25.90 |
|  | 53.92 | 0.00 | 8.57 | 37.75 | 44.37 | 53.37 | 25.27 | 67.28 | | |

**Impact of Task order** To explore the impact of different sequence order, we conduct an additional experiment using a different order of CoIN tasks, arranged alphabetically: GQA, Grounding, ImageNet, OCR-VQA, ScienceQA, TextVQA, VizWiz, and VQAV2. From the comparison results of LLaVA presented in Tab. 5, we observe that altering the task order inevitably influences the outcomes of each task. This effect occurs because the knowledge acquired from previous tasks can either benefit or hinder subsequent training. Furthermore, the final performance is also affected by the training sequence. Although the BWT of the alphabetic order is better than that of a random order, the overall result is still inferior to that achieved with a random order. After examining the overall performance throughout the training process, we observe that the results on the Grounding and ImageNet tasks are consistently inferior, thereby negatively impacting the overall performance.

Table 6: The results of LLaVA about **different instruction templates** are presented below.

| Type | Accuracy on Each Task | | | | | | | | Overall Results | |
|------|-----------|---------|----------|------|--------|-----------|-------|---------|-----|------|
|  | ScienceQA | TextVQA | ImageNet | GQA | VizWiz | Grounding | VQAV2 | OCR-VQA | MAA | BWT |
| Original | 82.45 | 49.99 | 96.05 | 56.40 | 55.45 | 31.27 | 62.20 | 57.08 | 32.97 | -32.62 |
|  | 21.26 | 28.74 | 10.25 | 36.78 | 32.45 | 0.83 | 42.50 | 57.08 | | |
| Diverse | 82.45 | 50.14 | 96.03 | 55.65 | 51.42 | 34.00 | 59.17 | 52.92 | 32.92 | -33.67 |
|  | 26.00 | 25.38 | 8.40 | 33.07 | 26.52 | 0.10 | 40.00 | 52.92 | | |
| 10Type | 81.65 | 51.99 | 97.00 | 61.30 | 54.10 | 39.20 | 68.15 | 64.65 | 38.37 | -31.75 |
|  | 54.84 | 35.46 | 9.80 | 38.70 | 12.95 | 0.82 | 46.80 | 64.65 | | |

**Impact of Instruction diversity** Furthermore, we note that some tasks in the CoIN dataset share similar instructions. This raises a pivotal question: does the type of instruction template impact the efficacy of continual instruction tuning? To investigate this issue, we devise two additional variants: 1) **Diverse**: Distinct instruction templates tailored to different tasks. 2) **10Type**: Randomly chosen from 10 distinct instruction templates. (Details can be found in the Appendix.) Tab. 6 presents the performance of task solving on these variants with LLaVA. Our comparative analysis reveals that merely changing to diverse templates has minimal impact on overall performance. However, randomly choosing from multiple distinct templates significantly enhances performance. The possible

reason is that with different templates, the model learns the true instructional intention within each task. Our findings suggest that instruction diversity can better mitigate the degeneration of instruction following, owing to increased robustness to varying instructions.

Table 7: The results of LLaVA about **different data volumes** are presented below.

| Volume | Accuracy on Each Task | | | | | | | | Overall Results | |
| | ScienceQA | TextVQA | ImageNet | GQA | VizWiz | Grounding | VQAV2 | OCR-VQA | MAA | BWT |
|---|---|---|---|---|---|---|---|---|---|---|
| 0.1 | 70.00 | 42.88 | 93.45 | 36.93 | 43.7 | 3.73 | 40.48 | 45.62 | 30.27 | -16.17 |
| | 53.71 | 32.62 | 5.38 | 33.50 | 36.98 | 2.85 | 36.77 | 45.62 | | |
| 0.2 | 69.86 | 46.86 | 94.38 | 44.98 | 44.15 | 4.81 | 32.55 | 52.10 | 30.33 | -19.89 |
| | 41.12 | 33.25 | 5.53 | 33.80 | 25.85 | 1.77 | 37.10 | 45.62 | | |
| 0.4 | 75.33 | 47.06 | 94.95 | 52.95 | 50.77 | 10.25 | 56.73 | 55.33 | 33.18 | -24.85 |
| | 49.96 | 23.60 | 7.22 | 36.12 | 33.05 | 0.09 | 39.20 | 55.33 | | |
| 0.6 | 78.09 | 47.65 | 95.85 | 55.93 | 53.08 | 10.00 | 59.17 | 46.33 | 31.47 | -32.57 |
| | 27.42 | 19.54 | 7.03 | 33.52 | 13.15 | 0.05 | 38.48 | 46.33 | | |
| 0.8 | 80.02 | 48.13 | 95.45 | 54.00 | 49.85 | 28.33 | 58.35 | 56.67 | 30.00 | -33.60 |
| | 11.74 | 16.94 | 8.85 | 32.62 | 35.50 | 0.00 | 39.67 | 56.67 | | |
| 1.0 | 82.45 | 49.99 | 96.05 | 56.40 | 55.45 | 31.27 | 62.20 | 57.08 | 32.97 | -32.62 |
| | 21.26 | 28.74 | 10.25 | 36.78 | 32.45 | 0.83 | 42.50 | 57.08 | | |

**Impact of data volume**    While there is ample knowledge retained in MLLMs, they require substantial instruction data for fine-tuning to enhance their ability to produce desired results. However, in practice, collecting high-quality data is costly. Several works [19, 51] have begun to study the impact of varying training data sizes on overall performance. In this work, to further explore the influence of the volume of instructions on continual instruction tuning, we conduct experiments to delve deeper into this investigation. To generate datasets of different volumes, we randomly select samples of each dataset from our benchmark, resulting in varying training data sizes, including 10%, 20%, 40%, 60%, and 80%. The experimental results of LLaVA are shown in Tab. 7. Overall, the performance exhibits an initial growth followed by a subsequent decline. This is possibly due to the fact that the model acquires more instruction following knowledge with the increase in size, as evidenced by the results of fine-tuning on each task growing with the volume increasing. However, the expansion in volume leads to the overriding of old knowledge by newly acquired knowledge, disrupting the balance between stability and plasticity and resulting in increased forgetting.

## 4.4   Example Analysis

1. TextVQA

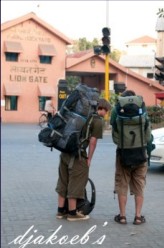

**User**: What is written on the front of that building? Reference OCR token:M,AVAL,LOOCKYARD,BOCK, ESTA,1735,.2,LION,GATE,djakoebs. Answer the question using a single word or phrase.
**Assistant**: Sunscreen
**Ground_truth**: lion gate

**Truth Alignment**: False
**Reasoning Capability**: 0

2. ScienceQA

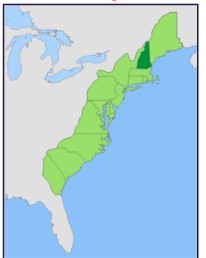

**User**: What is the name of the colony shown? A. Maryland B. New Hampshire C. Rhode Island D. Vermont. Answer with the option's letter from the given choices directly.
**Assistant**: New Hampshire
**Ground_truth**: B

**Truth Alignment** : False
**Reasoning Capability** : 10

3. GQA

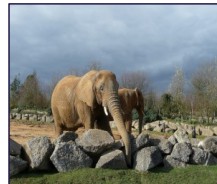

**User**: What animals are on the dirt? Answer the question using a single word or phrase.
**Assistant**: Elephants
**Ground_truth**: Elephants

**Truth Alignment** : True
**Reasoning Capability** : 10

4. ImageNet

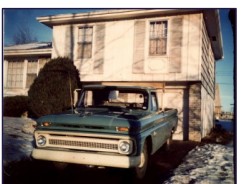

**User**: What is the object in the image? Answer the question using a single word or phrase.
**Assistant**: Car.
**Ground_truth**: Pickup

**Truth Alignment** : False
**Reasoning Capability** : 8

Figure 3: The illustration of test examples from LLaVA after training on the last task, *i.e.* OCR-VQA.

To further understand the difference between *Truth Alignment* and *Reasoning Capability*, we provide some examples after training on OCR-VQA in Fig. 3. In the first example, the model struggles to comprehend the content and produces an unrelated answer. Consequently, both *Truth Alignment* and *Reasoning Capability* evaluations result in a score of 0. In the second example from ScienceQA, the model comprehends the instruction and image, providing the correct answer "New Hampshire" instead of the intended instruction "B". Consequently, the outcome regarding *Truth Alignment* for this sample is incorrect. However, in reality, the output from the model encompasses all the reasoning knowledge necessary to solve this problem, earning a score of 10. As for the third example of the GQA, the output aligns with both the instruction intent and general knowledge, achieving the best result. The last example displays a result from ImageNet after training on OCR-VQA. We observed that after training on the last dataset, the model tends to give an answer "Car". While this response is considered incorrect since the ground truth answer is "Pickup", it is evident that the model has captured some knowledge contained in this sample pair. Fortunately, LLMs have analyzed the retained knowledge and output a suitable answer to indicate the degree of knowledge: 8.

## 5   Mixture-of-Experts benifits Continual Instruction Tuning

The Mixture-of-Experts (MoE) employs distinct experts to acquire various types of knowledge, akin to the expansion category of continual learning methods. Therefore, we bring the prevalent MoELoRA [36, 14] into CoIN to utilize experts to acquire distinct knowledge for different tasks to mitigate forgetting (Details are in Appendix). To validate the ability of MoELoRA to learn diverse knowledge and mitigate catastrophic forgetting, we conduct experiments with LLaVA by setting different expert numbers $N$ to the values of $\{2, 4, 8\}$, and report the quantitative results in Tab. 8 (Results of General Knowledge and other MLLMs are in Appendix). It is worth noting that the 1 expert in the MoELoRA method is equivalent to the vanilla LoRA fine-tuning method. Notably, the results demonstrate a consistent improvement across all metrics when the low-rank matrices of LoRA are divided into a greater number of experts. This trend can be attributed to the fact that enhanced specialization is achieved with more experts. Therefore, each distinct expert is capable of focusing on diverse instruction intent associated with specific tasks, effectively reducing interference.

Table 8: The results of LLaVA about **different numbers of experts** are presented below.

| Number | Accuracy on Each Task | | | | | | | | Overall Results | |
|---|---|---|---|---|---|---|---|---|---|---|
| | ScienceQA | TextVQA | ImageNet | GQA | VizWiz | Grounding | VQAV2 | OCR-VQA | MAA | BWT |
| Multi-task(1) | 56.77 | 49.35 | 95.55 | 56.65 | 53.90 | 30.09 | 59.50 | 55.65 | 57.18 | - |
| 1 | 82.45 | 49.99 | 96.05 | 56.40 | 55.45 | 31.27 | 62.20 | 57.08 | 32.97 | -32.62 |
| | 21.26 | 28.74 | 10.25 | 36.78 | 32.45 | 0.83 | 42.50 | 57.08 | | |
| 2 | 79.93 | 51.37 | 95.92 | 59.60 | 55.33 | 32.29 | 63.15 | 54.15 | 35.75 | -28.03 |
| | 47.77 | 31.67 | 10.75 | 37.10 | 40.98 | 1.44 | 43.65 | 54.15 | | |
| 4 | 80.35 | 52.21 | 96.25 | 59.62 | 58.05 | 34.47 | 64.40 | 62.73 | 40.24 | -26.57 |
| | 65.36 | 40.28 | 11.10 | 37.20 | 34.77 | 0.49 | 43.60 | 62.73 | | |
| 8 | 75.78 | 51.73 | 96.70 | 59.42 | 58.88 | 37.50 | 64.22 | 60.08 | 42.76 | -25.91 |
| | 63.09 | 38.63 | 10.50 | 37.38 | 43.62 | 0.59 | 43.15 | 60.08 | | |

**Comparison with Continual Methods**  To further investigate the effectiveness of our proposed method, we conduct experiments with other continual learning methods, including LwF and EWC. For EWC, we compute the Fisher matrix by randomly selecting 1,000 samples from each task and set the hyperparameter lambda to 0.1. For LwF, we choose to save 100 logits for each task to compute the distillation loss, the hyperparameter lambda is also set to 0.1. Further, since the experiments presented in Tab. 7 demonstrate that LLaVA achieves superior performance with a 40% data volume, we conduct the following experiments based on this setting and selected this model as the baseline. The experimental results based on LLaVA are illustrated in Tab. 9.

From the quantitative results shown below, we have several observations: (1). Our method consistently achieves the best final result, with improvements of 7.87% in MAA and 2.35% in BWT, respectively. (2). Our comparative analysis indicates that all approaches mitigate catastrophic forgetting. Notably, these methods primarily preserve knowledge in question-answering tasks but still experience forgetting on ImageNet and Grounding. Since EWC and LwF do not perform well

Table 9: The comparison with other continual learning methods based on LLaVA is presented below.

| Method | Accuracy on Each Task | | | | | | | | Overall Results | |
|---|---|---|---|---|---|---|---|---|---|---|
| | ScienceQA | TextVQA | ImageNet | GQA | VizWiz | Grounding | VQAV2 | OCR-VQA | MAA | BWT |
| Baseline | 75.33 | 47.06 | 94.95 | 52.95 | 50.77 | 10.25 | 56.73 | 55.33 | 33.18 | -24.85 |
| | 49.96 | 23.60 | 7.22 | 36.12 | 33.05 | 0.09 | 39.20 | 55.33 | | |
| LwF | 75.33 | 48.18 | 96.90 | 48.58 | 44.12 | 6.60 | 38.58 | 62.35 | 35.89 | -19.27 |
| | 63.14 | 39.60 | 8.90 | 34.83 | 14.53 | 2.48 | 40.67 | 62.35 | | |
| EWC | 75.28 | 48.37 | 96.83 | 42.77 | 44.25 | 8.65 | 60.27 | 61.02 | 40.36 | -17.94 |
| | 67.41 | 40.41 | 8.18 | 35.05 | 37.88 | 2.67 | 41.27 | 61.02 | | |
| MoELoRA | 75.85 | 49.05 | 93.95 | 56.53 | 48.70 | 25.57 | 61.9 | 55.35 | 41.05 | -22.50 |
| | 58.92 | 38.59 | 8.85 | 37.10 | 44.25 | 2.45 | 41.40 | 55.35 | | |

on the Grounding task, the forgetting in this task is less pronounced. (3). It is worth noting that under the 40% data volume setting, our method exhibits slightly more forgetting compared to other continual learning approaches. Upon further investigation, we find that this is to an enhancement in learning ability, as evidenced by improved performance on most tasks, particularly a 25.57% improvement on Grounding compared to other approaches. Consequently, our approach achieves better plasticity, achieving the best overall results. (4). The distributed training of large language models complicates the integration of EWC and LwF compared to our approach, which is designed based on the architecture and training paradigm of MLLMs. This poses a significant challenge that hinders the practical application of traditional continual learning approaches.

## 6   Limitations

Despite the positive contributions of this study, we acknowledge the following limitations: 1) **Model Size and Training Constraints**: This study only presents MLLMs ranging from 7 to 9 billion parameters. Due to computational limitations, we have not investigated larger models or employed a full fine-tuning strategy on our CoIN. 2) **Model Type**: Most MLLMs utilize LLaMA [57] as their language model, limiting the exploration of different model architectures. 3) **Task Diversity**: Currently, mainstream instruction tuning primarily focuses on image question answering tasks. Although we have incorporated classification and grounding tasks, it is crucial to explore the influence of a broader range of tasks.

## 7   Conclusion

This paper introduces a novel benchmark, Continual Instruction tuNing (CoIN), utilizing widely used vision-language datasets to investigate the behavior of Multimodal Large Language Models (MLLMs) on continual instruction tuning. CoIN encompasses 10 datasets spanning 8 tasks and transforms the data into an instruction-tuning format. Additionally, CoIN evaluates the MLLMs from truth alignment and reasoning capability. Experiments on CoIN explore the performance of MLLMs under different training orders, instruction types and data volumes. The results of these experiments show that the general knowledge maintained in MLLMs is robust for catastrophic forgetting, rather than instruction following. Based on this observation, we bring the MoELoRA into MLLMs to utilize different experts to learn the different tasks, effectively reducing catastrophic forgetting in MLLMs.

## Acknowledgments and Disclosure of Funding

This study is supported by grants from the National Natural Science Foundation of China (Grant No. 62122018, No. 62020106008, No. U22A2097, No. U23A20315), Kuaishou.

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

# Appendix

## A Datasheet

### A.1 Motivation

**Q: For what purpose was the dataset created?** This dataset is designed as a test-bed to investigate the behavior of Multimodal Large Language Models in continual instruction tuning. It specifically aims to address the lack of appropriate and diverse tasks for the instruction tuning of MLLMs.

**Q: Who created the dataset (e.g., which team, research group) and on behalf of which entity (e.g., company, institution, organization)?** The dataset was created by the authors, who are affiliated with the Center for Future Media Lab (CFM) located in the Computer Science and Engineering department at the University of Electronic Science and Technology of China (UESTC).

**Q: Who funded the creation of the dataset?** This work is supported by grants from the National Key Research and Development Program of China (2022YFC2009903/2022YFC2009900) and the National Natural Science Foundation of China (Grant No. 62122018, No. 62020106008, No. 61772116, No. 61872064).

**Q: Any other comments?** No.

### A.2 Composition

**Q: What do the instances that comprise the dataset represent (e.g., documents, photos, people, countries)?** Each instance represents a dialog between a human and an assistant, where the human asks a question based on the image, and the assistant answers the question based on its knowledge.

**Q: How many instances are there in total (of each type, if appropriate)?** As shown in Table 1, the dataset statistics are as follows:

- Grounding Task: 111,770 samples for training, 21,616 samples for testing.
- Classification Task: 117,715 samples for training, 4,600 samples for testing.
- VQAv2: 82,783 samples for training, 44,793 samples for testing.
- ScienceQA: 12,726 samples for training, 4,241 samples for testing.
- TextVQA: 34,602 samples for training, 5,000 samples for testing.
- GQA: 72,140 samples for training, 12,578 samples for testing.
- VizWiz: 20,523 samples for training, 8,000 samples for testing.
- OCR-VQA: 166,043 samples for training, 20,797 samples for testing.

**Q: Does the dataset contain all possible instances or is it a sample (not necessarily random) of instances from a larger set?** Most tuning datasets use the complete data from the original datasets, except for grounding and ImageNet. For grounding, we use only one annotation per image. For ImageNet, we randomly select 100 categories from the total 1000.

**Q: What data does each instance consist of?** Each instance consists of an image, an identifier, and a conversation list that includes instructions from the user and responses from the assistant.

**Q: Is there a label or target associated with each instance?** Each instance includes a value from the assistant that describes the ground truth of the output.

**Q: Is any information missing from individual instances?** No.

**Q: Are relationships between individual instances made explicit (e.g., users' movie ratings, social network links)?** No.

**Q: Are there recommended data splits (e.g., training, development/validation, testing)?** Yes, we have constructed the training and testing data. For validation, you can partition the training data as desired.

**Q: Are there any errors, sources of noise, or redundancies in the dataset?** No.

**Q: Is the dataset self-contained, or does it link to or otherwise rely on external resources (e.g., websites, tweets, other datasets)?** Since we construct the instruction from commonly used vision-language datasets, you need to download the images within these datasets, including GQA, TextVQA train,TextVQA test, ScienceQA, VizWiz train, VizWiz val, VizWiz test, OCR-VQA,ImageNet, COCO and the annotations for grounding RefCOCO,RefCOCO+,RefCOCOg.

**Q: Does the dataset contain data that might be considered confidential (e.g., data that is protected by legal privilege or by doctor–patient confidentiality, data that includes the content of individuals' non-public communications)?** No.

**Q: Does the dataset contain data that, if viewed directly, might be offensive, insulting, threatening, or might otherwise cause anxiety?** No.

**Q: Does the dataset relate to people?** No.

## A.3 Collection Process

**Q: How was the data associated with each instance acquired?** Our datasets are derived from publicly available and widely used vision-language datasets, which we transform into an instruction style using commonly employed templates.

**Q: What mechanisms or procedures were used to collect the data (e.g., hardware apparatuses or sensors, manual human curation, software programs, software APIs)?** We use a Python script for auto-labeling to generate instruction-style data.

**Q: If the dataset is a sample from a larger set, what was the sampling strategy (e.g., deterministic, probabilistic with specific sampling probabilities)?** Most tuning datasets use the complete data from the original datasets, except for grounding and ImageNet. For grounding, we use only one annotation per image. For ImageNet, we randomly select 100 categories from the total 1000.

**Q: Who was involved in the data collection process (e.g., students, crowdworkers, contractors) and how were they compensated (e.g., how much were crowdworkers paid)?** No crowdworkers were involved in the curation of the dataset. Open-source researchers and developers enabled its creation for no payment.

**Q: Over what timeframe was the data collected?** The whole instruction tuning data was generated in 2023.

**Q: Were any ethical review processes conducted (e.g., by an institutional review board)?** The source data of each task was collected through ethical review processes.

## A.4 Preprocessing/cleaning/labeling

**Q: Was any preprocessing/cleaning/labeling of the data done (e.g., discretization or bucketing, tokenization, part-of-speech tagging, SIFT feature extraction, removal of instances, processing of missing values)?** We utilize an auto-labeling preprocessing script to generate the instruction labels for the dataset. Apart from this, no additional preprocessing or labeling is performed.

**Q: Was the "raw" data saved in addition to the preprocessed/cleaned/labeled data (e.g., to support unanticipated future uses)?** Yes, we need to download the original images from the datasets upon which ours is based. The URLs for these datasets have been provided above.

**Q: Is the software that was used to preprocess/clean/label the data available?** Yes, we construct the tuning data by scripts.

## A.5 Uses

**Q: Has the dataset been used for any tasks already?** No.

**Q: Is there a repository that links to any or all papers or systems that use the dataset?** No.

**Q: What (other) tasks could the dataset be used for?** Although this dataset is created for continual instruction tuning, we can utilize the entire dataset for training a powerful assistant.

**Q: Is there anything about the composition of the dataset or the way it was collected and preprocessed/cleaned/labeled that might impact future uses?** No.

**Q: Are there tasks for which the dataset should not be used?** This dataset should not be used for commercial.

## A.6 Distribution

**Q: Will the dataset be distributed to third parties outside of the entity (e.g., company, institution, organization) on behalf of which the dataset was created?** Yes, the dataset will be open-source.

**Q: How will the dataset will be distributed (e.g., tarball on website, API, GitHub)?** The data is available through `https://huggingface.co/datasets/Zacks-Chen/CoIN`.

**Q: Will the dataset be distributed under a copyright or other intellectual property (IP) license, and/or under applicable terms of use (ToU)?** CC-4.0.

**Q: Have any third parties imposed IP-based or other restrictions on the data associated with the instances?** No.

**Q: Do any export controls or other regulatory restrictions apply to the dataset or to individual instances?** No.

## A.7 Maintenance

**Q: Who will be supporting/hosting/maintaining the dataset?** We will be hosting the dataset on huggingface.

**Q: How can the owner/curator/manager of the dataset be contacted (e.g., email address)?** The authors can be contacted via their emails mentioned in the paper. Issues can also be opened on our public GitHub repo.

**Q: Is there an erratum?** Not to the best of our knowledge.

**Q: Will the dataset be updated (e.g., to correct labeling errors, add new instances, delete instances)?** Maybe, we will add more diverse task into the CoIN.

**Q: Will older versions of the dataset continue to be supported/hosted/maintained?** Yes.

**Q: If others want to extend/augment/build on/contribute to the dataset, is there a mechanism for them to do so?** Not officially, but our benchmark code is open source and pull requests are welcome.

## B  Dataset details

The curated datasets are kept in JSON-files with the following keys:

- **id**: Identification for each instruction tuning sample.
- **image**: Image path.
- **conversation**: List of instructions and the answers from user and assistant.
  - **from**: Role of the instruction, human or gpt.
  - **value**: Instruction or answer details.

The constructed instruction training samples for each task are placed in **Instruction/Task_Name/train.json**, and testing samples are placed in **Instruction/Task_Name/test.json**. All code is accessible via the repository at `https://github.com/zackschen/CoIN`. In addition, all the training and testing instruction samples can be download from `https://huggingface.co/datasets/Zacks-Chen/CoIN`.

## C  Additional Experiments

**Impact of Backbone Size**   To evaluate the influence of different model sizes of backbone, we add a larger architecture to evaluate performance across different model sizes. We choose LLaVA-13B as the new backbone to conduct experiments on our proposed benchmark. The comparison of *Truth Alignment* and *Reasoning Capability* between the 13B and 7B models is presented in the table below.

Table 10: The results evaluating the *Truth Alignment* of LLaVA about **different model size** are presented below.

| Size | Accuracy on Each Task | | | | | | | | Overall Results | |
|------|-----------|---------|----------|------|--------|-----------|-------|---------|------|------|
|  | ScienceQA | TextVQA | ImageNet | GQA | VizWiz | Grounding | VQAV2 | OCR-VQA | MAA | BWT |
| 7B | 82.45 | 49.99 | 96.05 | 56.40 | 55.45 | 31.27 | 62.20 | 57.08 | 32.97 | -32.62 |
|  | 21.26 | 28.74 | 10.25 | 36.78 | 32.45 | 0.83 | 42.50 | 57.08 | | |
| 13B | 82.95 | 54.25 | 97.28 | 52.45 | 59.40 | 40.35 | 68.10 | 61.00 | 39.43 | -28.79 |
|  | 60.03 | 41.19 | 10.62 | 31.03 | 32.67 | 2.60 | 46.33 | 61.00 | | |

Table 11: The results evaluating the *Reasoning Capability* of LLaVA about **different model size** are presented below.

| Size | Accuracy on Each Task | | | | | | | | Overall Results | |
|------|-----------|---------|----------|------|--------|-----------|-------|---------|------|------|
|  | ScienceQA | TextVQA | ImageNet | GQA | VizWiz | Grounding | VQAV2 | OCR-VQA | MAA | BWT |
| 7B | 92 | 75 | 97 | 72 | 42 | 58 | 75 | 78 | 71.28 | -10.88 |
|  | | 82 | 74 | 55 | 56 | 47 | 52 | 58 | 78 | |
| 13B | 94 | 77 | 98 | 77 | 46 | 76 | 80 | 79 | 75.98 | -11.00 |
|  | 89 | 77 | 58 | 59 | 53 | 62 | 62 | 79 | | |

From these tables, we have the following observations: The learning ability increases with model size, evident in both Truth Alignment and Reasoning Capability, resulting in 39.43% and 75.98% in terms of MAA, respectively. In addition, the increase in model size mitigates catastrophic forgetting in Truth Alignment, resulting in a 3.83% improvement in terms of BWT. We believe this occurs because, with the increase in size, the model maintains a larger optimization space to learn new knowledge, allowing it to avoid overlapping with old knowledge. Finally, the observed decrease in forgetting for Truth Alignment and the increase in forgetting for Reasoning Capability suggest that the forgetting of Instruction Following is mitigating. This phenomenon indicates that increasing the architecture's size effectively mitigates the forgetting of the Instruction Following ability, which is valuable for the practical applications of MLLMs.

**Impact of rank of LoRA**   The text knowledge always exists when the parameters of the base LLM are frozen, which is consistent with our training setting (Section 3.1.2 in the paper). Therefore, any

forgetting primarily occurs in the multimodal knowledge acquired through the additional parameters introduced by LoRA which is very small compared with LLM. To examine this hypothesis further, we conduct additional experiments by increasing the rank of LoRA from 128 to 256. All experiments were conducted with a 40% data volume, as the experiments presented in Table 6 demonstrate that LLaVA achieves superior performance under this setting. The results are shown in the table below.

From Tab. 12, we first observe that performance improves as the rank increases, confirming that a higher number of trainable parameters enhances the model's ability to acquire new multimodal knowledge. Moreover, it is worth noting that knowledge forgetting is also reduced. This is likely because the additional parameters provide the model with sufficient optimization space to learn new multimodal information without overwriting previously utilized space.

Table 12: The results of LLaVA about **different rank of LoRA** are presented below.

| Rank | Accuracy on Each Task | | | | | | | | Overall Results | |
|---|---|---|---|---|---|---|---|---|---|---|
| | ScienceQA | TextVQA | ImageNet | GQA | VizWiz | Grounding | VQAV2 | OCR-VQA | MAA | BWT |
| 128 | 75.33 | 47.06 | 94.95 | 52.95 | 50.77 | 10.25 | 56.73 | 55.33 | 33.18 55.33 | -24.85 |
| | 49.96 | 23.60 | 7.22 | 36.12 | 33.05 | 0.09 | 39.2 | | | |
| 192 | 76.30 | 49.52 | 97.17 | 53.87 | 50.05 | 7.72 | 62.90 | 61.08 | 38.31 61.08 | -21.15 |
| | 68.82 | 40.63 | 8.72 | 35.70 | 30.45 | 2.95 | 41.08 | | | |
| 256 | 76.42 | 49.21 | 96.85 | 51.32 | 45.75 | 7.28 | 63.00 | 59.02 | 38.30 | -19.8 |
| | 69.13 | 38.51 | 7.58 | 36.1 | 33.83 | 3.85 | 41.42 | 59.02 | | |

# D   Examples of CoIN

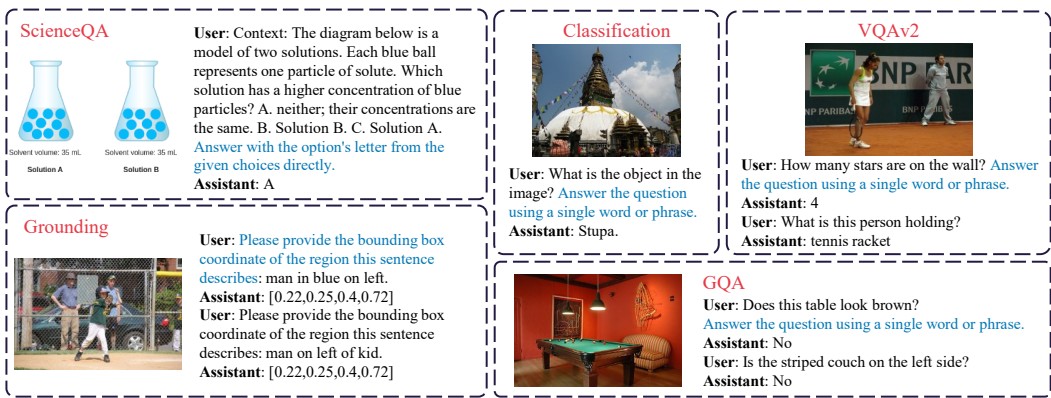

Figure 1: Examples of instruction tuning data in our proposed CoIN, which contains diverse visual understanding and perception tasks, such as classification, referring expression comprehension and image question answering.

To better show the constructed instruction data, we plot some examples with different tasks, as shown in Fig. 1. Blue text represents the instruction templates. We aim for the model to learn the capability of instruction following through these templates.

# E   *Truth Alignment* Comparison details

For the Image Question Answering task (including VQAv2, ScientQA, TextVQA, GQA, VizWiz, and OCR-VQA), we calculate the accuracy of predicting answers against ground truth, as in LLaVA [35]. In the classification task, the metric is computed by comparing predicted labels with real ones. For the referring expression comprehension task, we employ the widely used Intersection over Union (IoU) as the evaluation criterion to determine the success of the model's predictions. If the IoU of the predicted bounding box and the ground-truth bounding box is greater than 0.5, we consider the prediction to be correct.

## F  Experiments details

We conduct the experiments on LLaVA and Qwen-VL based on their official code. Following their official hyperparameters, we use the Adam optimizer with no weight decay and a cosine learning rate with a warmup ratio of 3%. During finetuning, gradient checkpointing is used to save GPU memory, and offloading is not used. BF16 and TF32 are enabled to achieve a balance between speed and precision. For MiniGPT-v2, we adapt the official code into $transformer$ training to utilize the $deepspeed$ for offloading frozen parameters to the CPU. In addition, we train all models with 8× 3090s with one epoch.

## G  More results about *Reasoning Capability*

Table 13: The results of LLaVA about **different task orders** are presented below.

| Order | Accuracy on Each Task | | | | | | | | Overall Results | |
|---|---|---|---|---|---|---|---|---|---|---|
| | ScienceQA | TextVQA | ImageNet | GQA | VizWiz | Grounding | VQAV2 | OCR-VQA | MAA | BWT |
| Random | 92 | 75 | 97 | 72 | 42 | 58 | 75 | 78 | 71.28 | -10.88 |
| | 82 | 74 | 55 | 56 | 47 | 52 | 58 | 78 | | |
| | GQA | Grounding | ImageNet | OCR-VQA | ScienceQA | TextVQA | VizWiz | VQAV2 | MAA | BWT |
| Alphabet | 77 | 75 | 98 | 81 | 80 | 75 | 48 | 79 | 68.88 | -3.00 |
| | 71 | 94 | 63 | 71 | 90 | 77 | 44 | 79 | | |

Tab. 13 presents the results regarding the *Reasoning Capability* of different task orders. From these comparison results, we observe that the *Reasoning Capability* follows a similar trend to *Truth Alignment*, with the performance of Random being better than Alphabet. Additionally, the forgetting of Alphabet is also milder compared to Random order, resulting in a -3.00% decrease in terms of BWT, indicating that task order impacts *Reasoning Capability*.

Table 14: The results of LLaVA about **different instruction templates** are presented below.

| Type | Accuracy on Each Task | | | | | | | | Overall Results | |
|---|---|---|---|---|---|---|---|---|---|---|
| | ScienceQA | TextVQA | ImageNet | GQA | VizWiz | Grounding | VQAV2 | OCR-VQA | MAA | BWT |
| Original | 92 | 75 | 97 | 72 | 42 | 58 | 75 | 78 | 71.28 | -10.88 |
| | 82 | 74 | 55 | 56 | 47 | 52 | 58 | 78 | | |
| Diverse | 92 | 76 | 97 | 71 | 47 | 61 | 71 | 80 | 72.54 | -9.62 |
| | 80 | 74 | 53 | 54 | 46 | 73 | 58 | 80 | | |
| 10Type | 92 | 76 | 98 | 76 | 41 | 74 | 79 | 83 | 74.21 | -10.00 |
| | 88 | 77 | 59 | 58 | 45 | 67 | 62 | 83 | | |

Tab. 14 reveals the performance of *Reasoning Capability* on different instruction templates. These comparisons also reveal that the impact of templates on *Reasoning Capability* mirrors that on *Truth Alignment*: simply switching to different templates has minimal effect on overall performance, but increased diversity in templates enhances the robustness of the model.

## H  Instruction diversity detail

We devise two additional instruction templates to investigate the impact of varying templates. The details of these templates are shown in Tab. 15.

## I  MoELoRA details

The Mixture-of-Experts (MoE) aims to activate a subset of parameters for each input, enabling a significant increase in model parameters without a corresponding increase in computational efforts. In commonly used transformer-based models, MoE typically transforms the feed-forward layer of each transformer block into an MoE layer [14, 36, 16]. This MoE layer comprises two modules: experts and gate function. The experts are several identical and independent feed-forward neural

Table 15: The list of instructions template for each task.

| Task | Original | Diverse | 10Type |
|---|---|---|---|
| ScienceQA | Answer with the option's letter from the given choices directly | Answer with the option's letter from the given choices directly | Answer with the option's letter from the given choices directly
Select the correct answer from the given choices and respond with the letter of the chosen option
Determine the correct option from the provided choices and reply with its corresponding letter
Pick the correct answer from the listed options and provide the letter of the selected option
Identify the correct choice from the options below and respond with the letter of the correct option
From the given choices, choose the correct answer and respond with the letter of that choice
Choose the right answer from the options and respond with its letter
Select the correct answer from the provided options and reply with the letter associated with it
From the given choices, select the correct answer and reply with the letter of the chosen option
Identify the correct option from the choices provided and respond with the letter of the correct option
From the given choices, pick the correct answer and respond by indicating the letter of the correct option |
| Grounding | Please provide the bounding box coordinate of the region this sentence describes | Please provide the bounding box coordinate of the region this sentence describes | Identify and provide the bounding box coordinates that match the description given in this sentence
Extract and provide the bounding box coordinates based on the region described in the sentence
Please provide the bounding box coordinate of the region this sentence describes
Find and provide the bounding box coordinates for the region described in the sentence
Provide the coordinates of the bounding box that correspond to the region described in the sentence
Give the bounding box coordinates as described in the sentence
Determine and provide the bounding box coordinates based on the description in the sentence
Identify and provide the coordinates of the bounding box described in the sentence
Provide the coordinates for the bounding box based on the region described in the sentence
Extract and provide the coordinates for the bounding box described in the sentence
Identify and give the coordinates of the bounding box as described by the sentence |
| GQA | Answer the question using a single word or phrase | Respond to the question briefly, using only one word or a phrase | Respond to the question with a single word or a short phrase
Respond to the question using only one word or a concise phrase
Answer the question with a single word or a brief phrase
Respond with one word or a short phrase
Provide your answer in the form of a single word or a concise phrase
Respond to the question with just one word or a brief phrase
Answer the question using a single word or a concise phrase
Provide your response using only one word or a short phrase
Respond to the question with a single word or a brief phrase
Respond to the question using just one word or a concise phrase
Answer the question with one word or a short phrase |
| ImageNet | Answer the question using a single word or phrase | Express your answer in a single word or a short, descriptive phrase | Express your answer in a single word or a short, descriptive phrase
Provide your answer using a single word or a brief phrase
Describe the content of the image using one word or a concise phrase
Respond to the question with a single word or a short, descriptive phrase
Classify the image content using only one word or a concise phrase
Give your answer in the form of a single word or a concise phrase
Use a single word or a short phrase to categorize the image content
Express your answer with one word or a short, descriptive phrase
Identify the type of content in the image using one word or a concise phrase
Summarize your response in a single word or a brief phrase
Use one word or a short phrase to classify the content of the image |
| OCR-VQA | Answer the question using a single word or phrase | Condense your answer for each question into a single word or concise phrase | Answer with the option's letter from the given choices directly
Select the correct answer from the given choices and respond with the letter of the chosen option
Determine the correct option from the provided choices and reply with its corresponding letter
Pick the correct answer from the listed options and provide the letter of the selected option
Identify the correct choice from the options below and respond with the letter of the correct option
From the given choices, choose the correct answer and respond with the letter of that choice
Choose the right answer from the options and respond with its letter
Select the correct answer from the provided options and reply with the letter associated with it
From the given choices, select the correct answer and reply with the letter of the chosen option
Identify the correct option from the choices provided and respond with the letter of the correct option
From the given choices, pick the correct answer and respond by indicating the letter of the correct option |
| TextVQA | Answer the question using a single word or phrase | Capture the essence of your response in a single word or a concise phrase | Answer the question with just one word or a brief phrase
Use one word or a concise phrase to respond to the question
Answer using only one word or a short, descriptive phrase
Provide your answer in the form of a single word or a brief phrase
Use a single word or a short phrase to respond to the question
Summarize your response in one word or a concise phrase
Respond to the question using a single word or a brief phrase
Provide your answer in one word or a short, descriptive phrase
Answer the question with a single word or a brief, descriptive phrase
Capture the essence of your response in one word or a short phrase
Capture the essence of your response in a single word or a concise phrase |
| VizWiz | Answer the question using a single word or phrase | Provide a succinct response with a single word or phrase | Answer the question using only one word or a concise phrase
Respond to the question using only one word or a concise phrase
Respond to the question with a single word or a brief phrase
Provide your answer using just one word or a short phrase
Respond with one word or a concise phrase
Answer the question with just one word or a brief phrase
Use a single word or a short phrase to answer the question
Provide your answer in the form of one word or a brief phrase
Reply to the question using one word or a concise phrase
Answer with a single word or a short phrase
Use one word or a brief phrase to answer the question |
| VQAv2 | Answer the question using a single word or phrase | Answer the question using a single word or phrase | Answer the question using a single word or phrase
Answer the question with a single word or a brief phrase
Use one word or a short phrase to respond to the question
Answer the question using just one word or a concise phrase
Provide your answer to the question using only one word or a brief phrase
Respond to the question with a single word or a short phrase Use a single word or phrase to answer the question
Provide an answer using only one word or a brief phrase
Answer the question succinctly with one word or a brief phrase
Answer the question with just one word or a short phrase
Respond to the question using a single word or a concise phrase |

networks, and the gate function models the probability distribution to govern the weights of outputs from these expert networks. Specifically, for an intermediate representation $x$ from the previous attention layer in models, the output of the MoE layer can be mathematically represented as follows:

$$h = \sum_{i=1}^{N} E_i(x)G(x)_i, \qquad (3)$$

where the $E_i(\cdot)$ and $G(\cdot)_i$ denote $i$-th expert and the gate function. In addition, the gate function can be written as follows:

$$G(h) = Softmax(hW_g), \qquad (4)$$

where $W_g$ is the trainable weight within gate function $G()$.

Our goal is to tackle the challenge of catastrophic forgetting in the continual instruction tuning of MLLMs. We are inspired by MoE, which employs distinct experts to acquire various types of knowledge, akin to the expansion category of continual learning methods. Therefore, we bring the prevalent method MoELoRA [36, 14] in CoIN to utilize experts to acquire distinct knowledge for different tasks to mitigate forgetting.

The MLLMs in CoIN are fine-tuned in a parameter-effective way, i.e Low-rank Adaptation (LoRA) [25]. LoRA uses two low-rank matrices with rank $r$ to update the knowledge and avoids changing the parameter of the learned model. Specifically, a certain transform feed-forward layer is parameterized with $W \in R^{d_{in} \times d_{out}}$, where $d_{in}$ and $d_{out}$ are the dimension of input and output, respectively. Two low-rank matrix $A \in R^{d_{in} \times r}$ and $B \in R^{r \times d_{out}}$ are used to learn extra knowledge with: $h = Wx + \frac{\alpha}{r} BAx$, where $x \in R^{d_{in}}$ and $h \in R^{d_{out}}$ denote the input and output vector, respectively. The rank $r$ controls the number of trainable matrices. In addition, the constant hyper-parameter $\alpha$ facilitates the tuning of rank $r$ [25].

To achieve the learning of diverse knowledge from different tasks, MoeLoRA proposes a set of experts to replace the LoRA matrices, denoted as $\{E_i\}_{i=1}^N$, where $N$ denotes the number of experts. Therefore, the original computation will change to:

$$h = Wx + \frac{\alpha}{r} \sum_{i=1}^N G_i E_i(x) = Wx + \frac{\alpha}{r} \sum_{i=1}^N G_i B_i A_i x, \tag{5}$$

where $G_i$ represents the gate function, which we will detail in the following paragraph. The matrices $A_i \in \mathbb{R}^{d_{in} \times \frac{r}{N}}$ and $B_i \in \mathbb{R}^{\frac{r}{N} \times d_{out}}$ represent the $i$-th expert of two low-rank matrices, each with a lower rank of $\frac{r}{N}$. With multiple experts in MoELoRA, the model can learn diverse task knowledge from different experts. Additionally, MoELoRA has the same number of trainable parameters as LoRA, indicating high efficiency.

Since there are many experts in each MoELoRA layer, the key is to create a suitable distribution of each expert to solve each task. As previously emphasized, to mitigate forgetting, the contribution of each expert should be tailored to specific tasks. Therefore, to regulate these contributions, a gate function is introduced. The gate function receives an input similar to the experts and outputs a contribution to choose suitable experts to solve the tasks. This computation is captured by the following equation:

$$G(x) = Softmax(xW_g), \tag{6}$$

where $W_g$ is the trainable weight within gate function $G(\cdot)$. To balance the scale of the output distribution, a softmax operation is applied to normalize the contribution weights. This output distribution is utilized to incorporate the varying percentage contributions of each expert, as outlined in Eq. 5. Ultimately, all the outputs are concatenated to form the final output for the next layer.

## J  Related Work

### J.1  Continual Learning

Recently, numerous methods have been proposed to mitigate catastrophic forgetting in the continual learning paradigm. These methods can be broadly categorized into three groups: *regularization-based*, *memory-based*, and *architecture-based* methods.

**Regularization-based** methods [52, 66, 1, 32, 7] focus on curing a continual learning network of its catastrophic forgetting by introducing an extra regularization term in the loss function. e.g, EWC[52] penalizes the changes of importance parameters when learning new tasks.

**Memory-based** methods [5, 49, 4, 39, 6] store previous samples or generate samples for replaying while learning a new task. Some methods [5, 49, 4, 4] use replayed samples from previous tasks to constrain the update of parameters when learning the new task. During training on a new task of EEC [2], reconstructed images from encoded episodes were replayed to avoid catastrophic forgetting.

**Architecture-based** methods [65, 27, 29, 43, 53] design new architecture modules to each task to prevent any possible forgetting. PNN [50] adds a network to each task and lateral connections to the network of the previous task while freezing previous task parameters. MNTDP [58] provides a learning algorithm to search the modules to combine with, where these modules represent atomic skills that can be composed to perform a certain task.

## J.2 Instruction Tuning

Instruction tuning is a promising approach to enable the pre-trained model to follow natural language instructions and improve their generalization performance to unseen tasks. Some methods [64, 10, 59, 63, 11, 34] use the existing vision-language datasets to create instruction tuning data by different templates. At the same time, some methods [35, 60, 24, 62, 56] use the existing vision datasets to generate instructions based on powerful LLMs (e.g GPT-4 [47]). LLaMA [57] observes that a very small amount of instructions improves the performance on MMLU [23], and further improves the ability of the model to follow instructions. LLaVA [35] leverages ChatGPT [46] and GPT-4 [47] for multimodal instruction-following data collection, based on the widely existing image-pair data. InstructBLIP [11] transforms 26 datasets into the instruction tuning format and groups them into 11 task categories for fine-tuning. To further enhance the instruction-following capacity, SPHINX [34] collects instruction data from a wide range of multi-modal tasks, and jointly fine-tune the model to learn a vision generalist, instead of a specialist for specific scenarios.

