# OpenReview forum: "CoIN: A Benchmark of Continual Instruction Tuning for Multimodel Large Language Models"
_NeurIPS.cc/2024/Datasets_and_Benchmarks_Track — NeurIPS 2024 Track Datasets and Benchmarks Poster_

### Official Review · Reviewer_FxCm · 2024-07-19
**This paper proposes a benchmark utilizing commonly used vision-language datasets and evaluates several baselines.**

**Rating:** 8
**Confidence:** 4
**Correctness:** In general, the claims are correct.
**Clarity:** The paper is well-written.

**Review:**

Please read as follows.

**Strengths:**

1. A diverse and comprehensive benchmark are constructed to investigate the behavior of MLLMs under continual fine-tuning. This serves as a cornerstone for exploring how to fine-tune these large models to adapt to new tasks amid the rapid development of MLLMs.
2. Two innovative evaluation metrics are proposed to comprehensively assess the behavior of MLLMs, considering different operational approaches between traditional CNNs or ViTs and MLLMs. Truth Alignment compares the model's outputs with ground truth, while Reasoning Capability, a creative metric, effectively evaluates the reasoning knowledge maintained in MLLMs using another LLM for assessment.
3. The experiments conducted in this paper are extensive and comprehensive. The results of these experiments reveal that in the instruction tuning of MLLMs, catastrophic forgetting primarily affects instruction following rather than retained knowledge. This finding can inspire further research to develop effective methods for preserving the instruction following ability of MLLMs.
4. Experiments with different instruction templates suggest that instruction diversity can better mitigate the degradation of instruction following, which is very valuable for the practical applications of MLLMs.
5. This benchmark, along with the corresponding code, have been released, enabling the reproducibility of results and providing the community with a valuable resource for further investigation of MLLMs.
6. The paper is well-written and easy to follow.

**Additional Feedback:**

No additional feedback. I have placed comments, suggestions, and questions spread out in the above.

**Documentation:**

This submission is very well documented and provides all the aspects asked for.

**Ethics:**

No ethical concerns.

**Opportunities For Improvement:**

This paper conducts extensive experiments, showcasing the performance of MLLMs comprehensively with the proposed metrics. It would be interesting to see how the model performs with a larger architecture, such as 13B, using this innovative benchmark.

**Relation To Prior Work:**

Yes, this paper has discussed the difference from previous work.

**Summary And Contributions:**

To investigate the behavior of Multimodal Large Language Models during sequential fine-tuning, this paper proposes a benchmark utilizing commonly used vision-language datasets and evaluates several baselines. Their experiments reveal that the model primarily loses its capability for instruction following rather than retained knowledge. Additionally, the authors conduct extensive ablation studies to examine the impact of various factors, including task order and data volume, and so on.

---

> ### Author Rebuttal · Authors · 2024-08-17
>
> We appreciate the detailed feedback you provided for our submission. We are encouraged by your acknowledgment that our “experiments are extensive and comprehensive”, “The finding can inspire further research”,  “Finds are valuable for the practical applications of MLLMs”, and "The paper is well-written and easy to follow".
> The following clarifications in response to your concerns are shown below:
>
> > **Larger Architecture** It would be interesting to see how the model performs with a larger architecture, such as 13B, using this innovative benchmark.
>
> Thank you for your suggestion to add a larger architecture to evaluate performance across different model sizes. We choose LLaVA-13B as the new backbone to conduct experiments on our proposed benchmark. The comparison of _Truth Alignment_ and _Reasoning Capability_ between the 13B and 7B models is presented in the table below.
>
> From these tables, we have the following observations:
> 1. The learning ability increases with model size, evident in both _Truth Alignment_ and _Reasoning Capability_, resulting in 39.43% and 75.98% in terms of MAA, respectively.
> 2. The increase in model size mitigates catastrophic forgetting in *Truth Alignment*, resulting in a 3.83% improvement in terms of BWT. We believe this occurs because, with the increase in size, the model maintains a larger optimization space to learn new knowledge, allowing it to avoid overlapping with old knowledge.
> 3. The observed decrease in forgetting for _Truth Alignment_ and the increase in forgetting for _Reasoning Capability_ suggest that the forgetting of _Instruction Following_ is mitigating. This phenomenon indicates that increasing the architecture's size effectively mitigates the forgetting of the _Instruction Following_ ability, which is valuable for the practical applications of MLLMs.
>
> Reflecting on our findings from this experiment, the final version of the manuscript will be revised accordingly.
>
> Table 1: The results evaluating the *Truth Alignment* ability are presented below.
>
> | Size |           |         | Accuracy |  of   |  Each  |   Task    |       |         | Overall | Results |
> | :--: | :-------: | :-----: | :------: | :---: | :----: | :-------: | :---: | :-----: | :-----: | :-----: |
> |      | ScienceQA | TextVQA | ImageNet |  GQA  | VizWiz | Grounding | VQAV2 | OCR-VQA |   MAA   |   BWT   |
> |  7B  |   82.45   |  49.99  |  96.05   | 56.40 | 55.45  |   31.27   | 62.20 |  57.08  |  32.97  | -32.62  |
> |      |   21.26   |  28.74  |  10.25   | 36.78 | 32.45  |   0.83    | 42.50 |  57.08  |         |         |
> | 13B  |   82.95   |  54.25  |  97.28   | 52.45 |  59.40  |   40.35   | 68.10  |   61.00    |  39.43  | -28.79  |
> |      |   60.03   |  41.19  |  10.62   | 31.03 | 32.67  |    2.60    | 46.33 |   61.00    |         |         |
>
> Table 2: The results evaluating the *Reasoning Capability* ability are presented below.
>
> | Size |           |         | Accuracy | of  |  Each  |   Task    |       |         | Overall | Results |
> | :--: | :-------: | :-----: | :------: | :-: | :----: | :-------: | :---: | :-----: | :-----: | :-----: |
> |      | ScienceQA | TextVQA | ImageNet | GQA | VizWiz | Grounding | VQAV2 | OCR-VQA |   MAA   |   BWT   |
> |  7B  |    92     |   75    |    97    | 72  |   42   |    58     |  75   |   78    |  71.28  | -10.88  |
> |      |    82     |   74    |    55    | 56  |   47   |    52     |  58   |   78    |         |         |
> | 13B  |    94     |   77    |    98    | 77  |   46   |    76     |  80   |   79    |  75.98  | -11.00  |
> |      |    89     |   77    |    58    | 59  |   53   |    62     |  62   |   79    |         |         |

---

> ### Author Rebuttal · Authors · 2024-08-30
>
> Dear reviewer, thank you again for taking the time to review our work. Let us know if you have any concerns left after our response. We would be happy to discuss any further questions and comments you may have. Please let us know if you have received our responses and if we have successfully addressed your concerns. Thank you once again for your feedback.

---

> ### Comment · Reviewer_FxCm · 2024-09-01
>
> The revision has addressed most of my concerns, especially by incorporating additional larger architecture models, which has greatly improved the paper's benchmark quality.
>
> I have also examined the comments from the other reviewers along with the authors' responses. The shared concerns about Qwen's effectiveness as an evaluator have been resolved through detailed experiments using GPT-4 and user studies, confirming its reliability.
>
> As a result, I have raised my rating.

---

### Official Review · Reviewer_zYZs · 2024-07-24
**Review for CoIN Benchmark**

**Rating:** 5
**Confidence:** 4
**Correctness:** Yes the claims made in the submission…
**Clarity:** Yes the paper was clear and easy-to-f…

**Review:**

See each section below.

**Strengths:**

1. The results include a lot of insights (e.g., multi-task model is not always the best, Zero-shot performance of VLMs are not optimal for all the tasks, MoE could retain the knowledge of each task & mitigate forgetting issue).

2. Problem setting is well formulated, and could be used as a valuable resource for future research on continual learning of VLMs.

**Additional Feedback:**

None

**Documentation:**

Yes it is included in the supplementary materials.

**Opportunities For Improvement:**

1. For assessing reasoning capability, the authors use Qwen-1.5-32B as the evaluator. Prior works such as [1] have shown that the best problem solving models (e.g., Llama-2-70B-Instruct) do not necessarily possess the ability to function as a good judge. Also, a good evidence to show that Qwen-1.5-32B could be used as a good judge would be to measure the correlation with human judgments. If this is too much, it is desirable to just use GPT-4 as a judge, which is conventionally done by a lot of papers.

2. Due to 1, I am not sure if the results and analysis based on Table 3 could be trusted.

3. The authors proposed MoE, a parameter-expansion method as a solution to mitigate the forgetting issue. In the CL literature, it is evident to add rehearsal methods or regularization methods as baselines.

**Relation To Prior Work:**

Yes the paper mentions and compares with prior works.

**Summary And Contributions:**

1. The authors propose a testbed called CoIN which is used to evaluate VLMs in a continual learning scenario.

2. Unlike the convention understanding that multitask training functions as a upper bound in a continual learning setting, the authors empirically show that sequential fine-tuning could outperform in certain cases for VLMs.

3. The authors show that MoE-based approach could mitigate the forgetting issue to some extent.

---

> ### Author Rebuttal · Authors · 2024-08-17
>
> Thank you for your valuable feedback. We are glad that you believe that our work reveals a lot of insights and formulates problem setting well, and could be used as a valuable resource for future research on the continual learning of VLMs.
> We respond to the concerns raised by you below. Please let us know what you think about our response and whether you would like further clarification.
>
>
> > **Evaluation Model**. Prior works such as [1] have shown that the best problem solving models (e.g., Llama-2-70B-Instruct) do not necessarily possess the ability to function as a good judge. Also, a good evidence to show that Qwen-1.5-32B could be used as a good judge would be to measure the correlation with human judgments. If this is too much, it is desirable to just use GPT-4 as a judge, which is conventionally done by a lot of papers. Due to 1, I am not sure if the results and analysis based on Table 3 could be trusted.
>
> Thank you for raising this issue. Indeed, some problem-solving models may not perform well as a good judge in some settings, so questioning whether a powerful problem-solving model is suitable for evaluating _Reasoning Capability_ is reasonable.
> Therefore, we conduct experiments by using GPT-4 model as an evaluator, and carry out a human judgment. The comparison results are shown below.
> Firstly, we incorporate the GPT-4 as an additional judge to assess the outputs, and the comparative results demonstrate that **Qwen's assessments are consistent with those of GPT-4**.
>
> Secondly, we conduct a human judgment to further validate the effectiveness of Qwen. **The study results also reveal similar trends with Qwen and GPT-4**. This measurement of the correlation between Qwen-32B and human judgments can be seen as strong evidence that Qwen-32B could serve as a good judge.
>
> Based on the comparison between Qwen and GPT-4, as well as human judgment, we believe that the results presented in Table 3 are accurate and reliable.
>
> | Evaluation Model |           |         | Accuracy | of  |  Each  |   Task    |       |         | Overall | Results |
> | :------------: | :-------: | :-----: | :------: | :-: | :----: | :-------: | :---: | :-----: | :-----: | :-----: |
> |                | ScienceQA | TextVQA | ImageNet | GQA | VizWiz | Grounding | VQAV2 | OCR-VQA |   MAA   |   BWT   |
> |    Qwen-32B    |    92     |   75    |    97    | 72  |   42   |    58     |  75   |   78    |  71.28  | -10.86  |
> |                |    82     |   74    |    55    | 56  |   47   |    52     |  58   |   78    |         |         |
> |     GPT-4      |    94     |   83    |    96    | 83  |   79   |    71     |  81   |   69    |  73.62  | -11.50  |
> |                |    80     |   83    |    65    | 67  |   62   |    70     |  68   |   69    |         |         |
> |   User Study   |    96     |   82    |    98    | 85  |   80   |    65     |  86   |   70    |  74.35  |  -8.13  |
> |                |    85     |   80    |    85    | 71  |   76   |    57     |  73   |   70    |         |         |
>
>
> >**Compare with other continual learning approaches.** In the CL literature, it is evident to add rehearsal methods or regularization methods as baselines.
>
> Thank you for your suggestions. We have incorporated two regularization methods, EWC and LwF, as new approaches into our analysis.
>
> Please refer to the Point 1 in general response for details.

---

> ### Author Rebuttal · Authors · 2024-08-30
>
> Dear reviewer, thank you again for taking the time to review our work. Let us know if you have any concerns left after our response. We would be happy to discuss any further questions and comments you may have. Please let us know if you have received our responses and if we have successfully addressed your concerns. Thank you once again for your feedback.

---

### Official Review · Reviewer_9Qeu · 2024-07-25
**A very timely and comprehensive benchmark for continual instruction tuning for multi-modality large language models**

**Rating:** 8
**Confidence:** 4
**Correctness:** Yes.
**Clarity:** Yes.

**Review:**

The quality, clarity, originality, and significance are all good enough to be accepted as a dataset and benchmark track paper.

**Strengths:**

To the best of my knowledge, this should be the first paper that can comprehensively benchmark different MLLMs for continual instruction tuning in a very organized manner. Recent state-of-the-art MLLMs like LLaVA, Qwen-VL, and MiniGPT-v2 are benchmarked in the present paper with extensive experiments and analysis, which will be treated as the cornerstone for future research. The research questions that the author has considered are relevant and interesting, and the takeaway messages are clear and sound. The considered baseline, i.e., MoELoRA, is also effective compared to the other methods benchmarked in the present paper.

**Additional Feedback:**

N/A.

**Documentation:**

Yes

**Ethics:**

N/A.

**Limitations:**

Please refer to Opportunities For Improvement for more details.

**Opportunities For Improvement:**

The main concern lies in the methods being compared. As a continual learning paper, the author should also benchmark other continual learning baselines, e.g., the regularization-based methods [d], as they can be readily implemented even for MLLM. Moreover, given that there have been many LoRA variants proposed since 2021, the author should also consider them for empirical comparison. Last but not least, the following paper [a, b, c] should be considered for comparison.

[a] InfLoRA: Interference-Free Low-Rank Adaptation for Continual Learning. CVPR 2024

[b] Boosting Continual Learning of Vision-Language Models via Mixture-of-Experts Adapters. CVPR 2024

[c] Omni-SMoLA: Boosting Generalist Multimodal Models with Soft Mixture of Low-rank Experts. CVPR 2024

[d] Overcoming catastrophic forgetting in neural networks. PNAS 2017

**Relation To Prior Work:**

Yes.

**Summary And Contributions:**

The author provides the first comprehensive benchmark for continual instruction tuning for multi-modality large language models. Relevant experimental protocols and analysis have been proposed tailored to the multi-modality large language models, where specific tasks, datasets, evaluation metrics, and instruction prompting templates have been considered. Three MLLMs have been chosen for benchmarking, and the author also proposes a simple baseline, i.e., MoELoRA, to mitigate the forgetting issue.

---

> ### Author Rebuttal · Authors · 2024-08-17
>
> We appreciate your recognition that our research questions are relevant and interesting, the takeaway messages are clear and sound and our benchmarks will be treated as the cornerstone for future research. We would like to clarify your concerns regarding comparisons with other continual learning baselines below.
>
> >**Compare with other continual learning approaches.** The author should also benchmark other continual learning baselines, e.g., the regularization-based methods [d], as they can be readily implemented even for MLLM.
>
> Thank you for your suggestions. We have incorporated EWC and LwF as new comparative approaches into our analysis.
>
> The experiment settings and results analysis are detailed in the general response. We kindly refer the reviewer to Point 1 for further information on this issue.
>
> In addition, thank you for your suggestion to add more LoRA variant methods. We will continue to update our work and include the results of these methods in future updates.

---

> ### Author Rebuttal · Authors · 2024-08-30
>
> Dear reviewer, thank you again for taking the time to review our work. Let us know if you have any concerns left after our response. We would be happy to discuss any further questions and comments you may have. Please let us know if you have received our responses and if we have successfully addressed your concerns. Thank you once again for your feedback.

---

### Official Review · Reviewer_XVFj · 2024-08-02

**Rating:** 5
**Confidence:** 3
**Correctness:** Yes
**Clarity:** Yes

**Review:**

Pros:
（1）The author has systematically uncovered the primary causes of catastrophic forgetting in MLLMs by collecting a comprehensive dataset and conducting tests on truth alignment and reasoning capabilities. The findings reveal that catastrophic forgetting is mainly attributed to a decline in following instructions rather than a deterioration in reasoning knowledge.
（2） Through detailed analysis, the author has highlighted the significant effects of data volume, instruction diversity, and task order during training on model overfitting. These insights are immensely beneficial for subsequent academic research.
Cons:
（1）The author has primarily compiled existing benchmarks, which does not sufficiently contribute to the development of new benchmarks. Additionally, when assessing reasoning capabilities, the author has used other open-source large models for evaluation, which could introduce a bias compared to actual results. It would be beneficial if the author could confirm whether they conducted a sample and manually verified the accuracy of these evaluations.
（2）It appears evident that the model has not forgotten its learned knowledge. Many MLLMs are trained without updating the LLM’s parameters. The knowledge (parameters) learned internally by the LLM still exists (in other words, even after removing these additional modules, the model retains its text capabilities). I think the author should attempt to update the overall parameters using multimodal data or increase the rank of LoRA to test if using solely multimodal data leads to knowledge forgetting in the model.
（3）Although the method proposed by the author offers some mitigation of this phenomenon, the improvement is minimal, as compared from Tables 2 and 7. The enhancements in grounding and performance on ImageNet are not as effective as anticipated. I suggest that in the comparison in Table 7, the author should also include a comparison with multi-task settings to provide a more detailed analysis.

**Strengths:**

refer to the above

**Additional Feedback:**

NA

**Documentation:**

There is sufficient detail to support reproducibility.

**Limitations:**

For the evaluation of reasoning capabilities, the author should consider testing a more diverse array of models. Additionally, manual sampling to verify the accuracy of model evaluations is recommended to ensure the reliability of the results.

**Opportunities For Improvement:**

refer to the above

**Relation To Prior Work:**

The authors are clearly discussed how this work differs from previous contributions

**Summary And Contributions:**

This paper introduces the CoIN benchmark to assess Multimodal Large Language Models (MLLMs) using vision-language datasets across various tasks. The evaluation focuses on truth alignment and reasoning abilities, highlighting that MLLMs retain general knowledge but struggle with following specific instructions due to catastrophic forgetting. To address this, the MoELoRA framework is integrated into MLLMs, employing task-specific experts to enhance their performance and reduce forgetting.

---

> ### Author Rebuttal · Authors · 2024-08-17
>
> **Rebuttal [1/3]**
>
> Thank you for your review and thoughtful comments. We are glad that you find the insights from our work to be immensely beneficial for subsequent academic research.
> We address each of your points of feedback below.
>
> > **Compiling existing benchmarks**. The author has primarily compiled existing benchmarks, which does not sufficiently contribute to the development of new benchmarks.
>
> First, the main goal of our paper is to evaluate the effectiveness of MLLMs within the continual learning paradigm. However, as **there was no existing benchmark** for this purpose, we select **public, diverse, and high-quality datasets** to create our instruction tuning data. In addition, public datasets are also commonly used in many studies [1,2,3] to train effective MLLMs and create benchmarks.
>
> Second, our benchmark conducts comprehensive experiments and detailed analysis to gain valuable insights, as recognized by other reviewers:
> 1. **Reviewer 9Qeu** notes that our work '**comprehensively benchmarks** different MLLMs for continual instruction tuning in a very organized manner,' and that the research questions 'are relevant and interesting, with **clear and sound takeaway messages.**'
> 2. **Reviewer FxCm** highlights that our experimental setup 'proposes two innovative evaluation metrics,' and that the findings '**can inspire further research and are valuable** for the practical applications of MLLMs.'
> 3. **Reviewer FxCm** believes that this benchmark, along with our released instructions and code will provide the community with **essential resources for further investigation** into MLLMs.
> 4. **Reviewer zYZs**  believes that our work could be used as a **valuable resource** for future research on continual learning of VLMs.
>
> > **Evaluation Model**. When assessing reasoning capabilities, the author has used other open-source large models for evaluation, which could introduce a bias compared to actual results. It would be beneficial if the author could confirm whether they conducted a sample and manually verified the accuracy of these evaluations.
>
> Thank you for your suggestion. Indeed, there my be bias by introducing an open-source model as an evaluator, raising an open question about whether this bias affects the assessment of *Reasoning Capability*. To examine the effect of the bias, we conduct experiments, using a powerful closed large language model as an evaluator, and also carry out a user study. The comparison results are shown below.
>
> Firstly, many works [3,4] have utilized GPT-4 to evaluate the quality of samples; therefore, we also employ GPT-4 to evaluate the outputs with the same prompts.
> The comparison with Qwen reveals that **the overall trends about the assessment of *Reasoning Capability* are consistent**.
>
> Secondly, we randomly sample model outputs for each task and gather feedback from researchers in the field of artificial intelligence, asking them to score the outputs by using the same prompts with both GPT-4 and Qwen-32B.
> The overall trends of the user study results are similar to those of Qwen-32B, validating that Qwen-32B could serve as a reliable evaluator.
>
> In summary, while employing open-source models as evaluators may introduce some bias, the impact under our setting is minimal. Qwen remains effective in assessing the retention and forgetting of *Reasoning Capability*.
>
> | Evaluation Model |           |         | Accuracy | of  |  Each  |   Task    |       |         | Overall | Results |
> | :------------: | :-------: | :-----: | :------: | :-: | :----: | :-------: | :---: | :-----: | :-----: | :-----: |
> |                | ScienceQA | TextVQA | ImageNet | GQA | VizWiz | Grounding | VQAV2 | OCR-VQA |   MAA   |   BWT   |
> |    Qwen-32B    |    92     |   75    |    97    | 72  |   42   |    58     |  75   |   78    |  71.28  | -10.86  |
> |                |    82     |   74    |    55    | 56  |   47   |    52     |  58   |   78    |         |         |
> |     GPT-4      |    94     |   83    |    96    | 83  |   79   |    71     |  81   |   69    |  73.62  | -11.50  |
> |                |    80     |   83    |    65    | 67  |   62   |    70     |  68   |   69    |         |         |
> |   User Study   |    96     |   82    |    98    | 85  |   80   |    65     |  86   |   70    |  74.35  |  -8.13  |
> |                |    85     |   80    |    85    | 71  |   76   |    57     |  73   |   70    |         |         |
>
>
> [1]. Visual Instruction Tuning, NeurIPS, 2024.
>
> [2]. ImageBind: One Embedding Space To Bind Them All, CVPR, 2023.
>
> [3]. PIXIU: A Large Language Model, Instruction Data and Evaluation Benchmark for Finance, NeurIPS, 2023.
>
> [4]. Textbooks Are All You Need, 2023.
>
> [5]. G-EVAL: NLG Evaluation using GPT-4 with Better Human Alignment, 2023.

---

> ### Author Rebuttal · Authors · 2024-08-17
>
> **Rebuttal [2/3]**
>
> >  **Increasing the rank of LoRA**. I think the author should attempt to update the overall parameters using multimodal data or increase the rank of LoRA to test if using solely multimodal data leads to knowledge forgetting in the model.
>
> Thank you for your valuable suggestion. As you mentioned, the text knowledge still exists when the parameters of the base LLM are frozen, which is consistent with our training setting (Section 3.1.2 in the paper). Therefore, any forgetting primarily occurs in the multimodal knowledge acquired through the additional parameters introduced by LoRA which is very small compared with LLM. To examine this hypothesis further, we conduct additional experiments by increasing the rank of LoRA from 128 to 256.
> All experiments were conducted with a 40% data volume, as the experiments presented in Table 6 demonstrate that LLaVA achieves superior performance under this setting. The results are shown in the table below.
>
> From this table, we first observe that performance improves as the rank increases, confirming that a higher number of trainable parameters enhances the model's ability to acquire new multimodal knowledge.
> Moreover, it is worth noting that knowledge forgetting is also reduced. This is likely because the additional parameters provide the model with sufficient optimization space to learn new multimodal information without overwriting previously utilized space.
>
> | Rank |           |         | Accuracy |  of   |  Each  |   Task    |       |         | Overall | Results |
> | :--: | :-------: | :-----: | :------: | :---: | :----: | :-------: | :---: | :-----: | :-----: | :-----: |
> |      | ScienceQA | TextVQA | ImageNet |  GQA  | VizWiz | Grounding | VQAV2 | OCR-VQA |   MAA   |   BWT   |
> | 128  |   75.33   |  47.06  |  94.95   | 52.95 | 50.77  |   10.25   | 56.73 |  55.33  |  33.18  | -24.85  |
> |      |   49.96   |  23.60  |   7.22   | 36.12 | 33.05  |   0.09    | 39.2  |  55.33  |         |         |
> | 192  |   76.30   |  49.52  |  97.17   | 53.87 | 50.05  |   7.72    | 62.9  |  61.08  |  38.31  | -21.15  |
> |      |   68.82   |  40.63  |   8.72   | 35.7  | 30.45  |   2.95    | 41.08 |  61.08  |         |         |
> | 256  |   76.42   |  49.21  |  96.85   | 51.32 | 45.75  |   7.28    | 63.00 |  59.02  |  38.30  |  -19.8  |
> |      |   69.13   |  39.51  |   7.58   | 36.1  | 33.83  |   3.85    | 41.42 |  59.02  |         |         |

---

> ### Author Rebuttal · Authors · 2024-08-17
>
> **Rebuttal [3/3]**
>
> > **Improvement is minimal.** Although the method proposed by the author offers some mitigation of this phenomenon, the improvement is minimal, as compared from Tables 2 and 7. The enhancements in grounding and performance on ImageNet are not as effective as anticipated.
>
> Thank you for bringing up the discussion of our method.
> Firstly, after incorporating EWC and LwF into our benchmark (as shown in the general response), we observe that **they also achieve minimal improvements on Grounding and ImageNet**. This highlights that this is a significant challenge for the entire continual learning community.
> Regarding our MoELoRA, we conjecture that splitting LoRA into different experts may cause them to focus on acquiring various types of knowledge within similar tasks, as evidenced by the mitigation of forgetting across all QA tasks.
>
> Secondly, we proposed MoELoRA, a method resembling model expansion, **based on the architecture and training paradigm of MLLMs.** While its improvements on tasks with substantial gaps (e.g., grounding and ImageNet) are less pronounced, it effectively mitigates forgetting in question-answering tasks. These results validate the potential of our approach, marking a meaningful step forward in continual learning research within the field of MLLMs.
>
> >  **Detailed analysis in Table 7.** I suggest that in the comparison in Table 7, the author should also include a comparison with multi-task settings to provide a more detailed analysis.
>
> We provide a more detailed analysis by incorporating the multi-task results into Table 7.
> Totally,  the results demonstrate a consistent improvement when the low-rank matrices of LoRA are divided into a greater number of experts.
>
> Additionally, the mitigation of forgetting is primarily focusing on question-answering tasks. After carefully examining the outputs for Grounding and ImageNet tasks, we find that failures in instruction alignment still play a significant role in the observed performance issues.
>
> Lastly, it is noteworthy that the tasks presented earlier in the sequence generally suffer less from forgetting compared to later tasks. This suggests that earlier tasks may utilize most of the available experts, resulting in limited improvement for subsequent tasks.
>
> |    Number     |           |         | Accuracy |  of   |  Each  |   Task    |       |         | Overall | Results |
> | :-----------: | :-------: | :-----: | :------: | :---: | :----: | :-------: | :---: | :-----: | :-----: | :-----: |
> |               | ScienceQA | TextVQA | ImageNet |  GQA  | VizWiz | Grounding | VQAV2 | OCR-VQA |   MAA   |   BWT   |
> | Multi-task(1) |   56.77   |  49.35  |  95.55   | 56.65 | 53.90  |   30.09   | 59.50 |  55.65  |  57.18  |    -    |
> |       1       |   82.45   |  49.99  |  96.05   | 56.40 | 55.45  |   31.27   | 62.20 |  57.08  |  32.97  | -32.62  |
> |               |   21.26   |  28.74  |  10.25   | 36.78 | 32.45  |   0.83    | 42.50 |  57.08  |         |         |
> |       2       |   79.93   |  51.37  |  95.92   | 59.6  | 55.33  |   32.29   | 63.15 |  54.15  |  35.75  | -28.03  |
> |               |   47.77   |  31.67  |  10.75   | 37.10 | 40.98  |   1.44    | 43.65 |  54.15  |         |         |
> |       4       |   80.35   |  52.21  |  96.25   | 59.62 | 58.05  |   34.47   | 64.40 |  62.73  |  40.24  | -26.57  |
> |               |   65.36   |  40.28  |  11.10   | 37.20 | 34.77  |   0.49    | 43.60 |  62.73  |         |         |
> |       8       |   75.78   |  51.73  |  96.70   | 59.42 | 58.88  |   37.50   | 64.22 |  60.08  |  42.76  | -25.91  |
> |               |   63.09   |  38.63  |   10.5   | 37.38 | 43.62  |   0.59    | 43.15 |  60.08  |         |         |

---

> ### Author Rebuttal · Authors · 2024-08-30
>
> Dear reviewer, thank you again for taking the time to review our work. Let us know if you have any concerns left after our response. We would be happy to discuss any further questions and comments you may have. Please let us know if you have received our responses and if we have successfully addressed your concerns. Thank you once again for your feedback.

---

### Author Response · Authors · 2024-08-17
**General Response**

Dear reviewers,

We thank the reviewers for their time, constructive comments, thoughtful questions, and positive feedback.

It's encouraging to note the contributions of our work highlighted by reviewers: Our benchmark collects a **comprehensive dataset** (XVFj) to create the **first comprehensive benchmark** (9Qeu, FxCm) for continual instruction tuning in a **very organized manner** (9Qeu). We propose **two innovative evaluation metrics** to assess the behavior of MLLMs (FxCm). **Extensive and comprehensive experiments** and **detailed analysis** are conducted in this paper (XVFj, 9Qeu, FxCm), leading to **a lot of insights** that are considered a **valuable resource for future research** on the continual learning of MLLMs (XVFj, zYZs, 9Qeu, FxCm). Our proposed method is also noted to be **effective in retaining knowledge and mitigating the forgetting issue** (9Qeu).

Based on your feedback, we have further enriched the benchmarks with additional experiments and baselines, validated the use of Qwen as an evaluator, and addressed all the points you raised. We believe the submission is much stronger as a result.

Here we address the common question raised by the reviewers:

## 1. Compare with other continual learning approaches.

We have incorporated EWC and LwF as new comparative approaches into our analysis.
For EWC, we compute the Fisher matrix by randomly selecting 1,000 samples from each task and set the hyperparameter lambda to 0.1.
For LwF, we choose to save 100 logits for each task to compute the distillation loss, the hyperparameter lambda is also set to 0.1.
Since the experiments presented in Table 6 demonstrate that LLaVA achieves superior performance with a 40% data volume, we conduct the following experiments based on this setting and selected this model as the baseline.

From the quantitative results shown below, we have the following observations:

(1). Our method consistently **achieves the best final result**, with improvements of 7.87% in MAA and 2.35% in BWT, respectively.

(2). Our comparative analysis indicates that all approaches mitigate catastrophic forgetting. Notably, these methods primarily preserve knowledge in question-answering tasks but still experience forgetting on ImageNet and Grounding. Since EWC and LwF do not perform well on the Grounding task, the forgetting in this task is less pronounced.

(3). It is worth noting that under the 40% data volume setting, our method exhibits slightly more forgetting compared to other continual learning approaches. Upon further investigation, we find that this is due to an **enhancement in learning ability**, as evidenced by improved performance on most tasks, particularly a 25.57% improvement on Grounding compared to other approaches. Consequently, our approach achieves better plasticity, achieving the best overall results.

(4). The distributed training of large language models **complicates the integration** of EWC and LwF compared to our approach, **which is designed based on the architecture and training paradigm of MLLMs**. This poses a significant challenge that hinders the practical application of traditional continual learning approaches.

| Approach |           |         | Accuracy |  of   |  Each  |   Task    |       |         | Overall | Results |
| :------: | :-------: | :-----: | :------: | :---: | :----: | :-------: | :---: | :-----: | :-----: | :-----: |
|          | ScienceQA | TextVQA | ImageNet |  GQA  | VizWiz | Grounding | VQAV2 | OCR-VQA |   MAA   |   BWT   |
| Baseline |   75.33   |  47.06  |  94.95   | 52.95 | 50.77  |   10.25   | 56.73 |  55.33  |  33.18  | -24.85  |
|          |   49.96   |  23.60  |   7.22   | 36.12 | 33.05  |   0.09    | 39.20 |  55.33  |         |         |
|   LwF    |   75.33   |  48.18  |  96.90   | 48.58 | 44.12  |   6.60    | 38.58 |  62.35  |  35.89  | -19.27  |
|          |   63.14   |  39.6   |   8.90   | 34.83 | 14.53  |   2.48    | 40.67 |  62.35  |         |         |
|   EWC    |   75.28   |  48.37  |  96.83   | 42.77 | 44.25  |   8.65    | 60.27 |  61.02  |  40.36  | -17.94  |
|          |   67.41   |  40.41  |   8.18   | 35.05 | 37.88  |   2.67    | 41.27 |  61.02  |         |         |
| MoELoRA  |   75.85   |  49.05  |  93.95   | 56.53 |  48.7  |   25.57   | 61.9  |  55.35  |  41.05  | -22.50  |
|          |   58.92   |  38.59  |   8.85   | 37.10 | 44.25  |   2.45    | 41.4  |  55.35  |         |         |

We now respond to each reviewer individually.

---

### Decision · Program_Chairs · 2024-09-26

**Decision:**

Accept (Poster)

**Comment:**

This paper provides a comprehensive benchmark for continual instruction tuning for multi-modality large language models. All reviewers agree the topic is important. The main concerns of the reviewers lie in the evaluation against existing continual learning approaches, the choice of evaluation model and backbone model size. The rebuttal of the authors has addressed these issues with extensive experiments. Hence, I believe the paper is now of good shape.